# Offline Constrained Multi-Objective Reinforcement Learning via Pessimistic Dual Value Iteration

**Runzhe Wu**
Shanghai Jiao Tong University
`runzhe@sjtu.edu.cn`

**Yufeng Zhang**
Northwestern University
`yufengzhang2023@u.northwestern.edu`

**Zhuoran Yang**
Princeton University
`zy6@princeton.edu`

**Zhaoran Wang**
Northwestern University
`zhaoranwang@gmail.com`

## Abstract

In constrained multi-objective RL, the goal is to learn a policy that achieves the best performance specified by a multi-objective preference function under a constraint. We focus on the offline setting where the RL agent aims to learn the optimal policy from a given dataset. This scenario is common in real-world applications where interactions with the environment are expensive and the constraint violation is dangerous. For such a setting, we transform the original constrained problem into a primal-dual formulation, which is solved via dual gradient ascent. Moreover, we propose to combine such an approach with pessimism to overcome the uncertainty in offline data, which leads to our Pessimistic Dual Iteration (PEDI). We establish upper bounds on both the suboptimality and constraint violation for the policy learned by PEDI based on an arbitrary dataset, which proves that PEDI is provably sample efficient. We also specialize PEDI to the setting with linear function approximation. To the best of our knowledge, we propose the first provably efficient constrained multi-objective RL algorithm with offline data without any assumption on the coverage of the dataset.

## 1 Introduction

There has been increased interest in multi-objective RL in recent years. Compared with traditional single-objective RL, the goal of the multi-objective one depends on a preference function, which takes the multiple objectives as input and outputs a scalar. The multi-objective optimization problems are usually constrained, as otherwise, it may cause danger or malfunction in applications. For example, consider a home automation system that helps humans monitor and control home attributes which can be regarded as multi-objectives. Users at different times may value different aspects of its services. Some may think highly of lighting at night, while others may concern the climate. We can formulate the users' preference as a preference function on the multiple objectives. Therefore, the system is dealing with a multi-objective optimization problem. However, the system cannot optimize this problem without any constraints, which might go against human's will, such as generating extreme climate inside a house or performing unacceptably energy-consuming operations.

We formulate the constrained multi-objective Markov decision process (CMOMDP), which is similar to the constrained Markov decision process (CMDP) (Altman, 1999). The difference is that the objectives are multiple and constraints can be nonlinear. We aim to minimize the value of a preference function, which takes multiple objectives as input and outputs a scalar.

35th Conference on Neural Information Processing Systems (NeurIPS 2021).

Most existing RL methods assume full accessibility of the environment for the agent, which tends to be impractical as the exploration could be expensive (Gottesman et al., 2019) and dangerous (Shalev-Shwartz et al., 2016). Hence, we consider the offline case where the agent has only a historical dataset collected a priori and has no further interactions with the environment. This setting is common and embodied in various scenarios such as healthcare (Chakraborty and Murphy, 2014) and auto-driving (Sun et al., 2020). However, offline RL is less understood theoretically (Levine et al., 2020) than online RL, and how to maximally exploit the dataset remains unclear.

In this paper, we study the offline CMOMDP. The challenges are threefold:

(i) Different from CMDPs, the nonlinearity of the preference function and constraints of CMOMDPs makes the analysis challenging. Moreover, constraints can be neither convex nor concave in the policy, which means the optimization problems are usually non-convex. It brings great difficulty to designing a provably efficient algorithm.

(ii) Since further interaction with the environment is prohibited, the dataset's quality is not guaranteed. The data collected by the experimenter may not sufficiently cover the trajectories induced by the optimal policy. Therefore, the information of the optimal policy could be limited, which probably makes the suboptimality and constraint violation arbitrarily large.

(iii) As the CMOMDP can be considered a generalization of CMDP (see the reduction in Appendix C), any difficulties emerging in the CMDP will arise here too. For example, due to constraints, the Bellman optimality equation may not hold anymore. Therefore, most existing offline RL algorithms based on dynamic programming are inapplicable.

Challenge (i) is inherent in CMOMDPs, and certain requirements on the preference function and constraints are inevitable. Hence, We suppose they satisfy some conditions. For instance, a geometric analog of Slater's condition, which is usually assumed in CMDPs, is imposed. To tackle challenge (ii), existing works have put a great effort. One possible principle is pessimism, which applies penalties to ensure pessimistic estimation. This method is successful in ordinary RL, and we extend it to CMOMDPs. We also note that we only impose a minimal assumption on the offline dataset's compliance in our analysis. For challenge (iii), we apply the convex conjugate and Fenchel's duality to transform the problem into a primal-dual formulation.

In summary, our work answers the following question:

*Is it possible to develop a provably efficient offline algorithm for constrained multi-objective reinforcement learning with minimal assumptions on the dataset?*

We propose the Pessimistic Dual Iteration (PEDI) algorithm. Theoretical contributions are as follows:

(i) By transforming the original constrained optimization problem of CMOMDPs into a primal-dual formulation via convex conjugate and duality, we develop the algorithm for general CMOMDPs with offline dataset, which iterates in a dual gradient ascent manner. Then, we instantiate the algorithm for linear kernel CMOMDPs, which is a large class of CMOMDPs that includes the tabular case.

(ii) We show in Appendix F the significance of pessimism in offline CMOMDPs. To summarize, we decompose the discrepancy between a value function and the optimal one into spurious correlation, intrinsic uncertainty, and optimization error, among which the spurious correlation is the most difficult to control. However, by maintaining pessimistic estimates of the value functions, PEDI eliminates the spurious correlation.

(iii) We establish theoretical guarantees for PEDI. Specifically, we demonstrate that two metrics, the suboptimality and the constraint violation, can be bounded from above by the optimization error of $\mathcal{O}(1/\sqrt{K})$ and the intrinsic uncertainty, where $K$ is the number of iterations. When specialized to linear kernel CMOMDPs, the upper bound of suboptimality matches the information-theoretic lower bound up to the optimization error and multiplicative factors of constants related to the CMOMDP, which suggests the near-optimality of PEDI.

(iv) We show that the error of PEDI is data-dependent, i.e., it depends on how well the dataset covers the trajectories induced by the optimal policy. When the trajectories are assumed further to be sufficiently covered, the suboptimality and constraint violation are $\widetilde{\mathcal{O}}(1/\sqrt{N})$ where $N$ is the number of trajectories in the dataset.

To the best of our knowledge, we are the first to propose a provably efficient offline algorithm that considers the constrained multi-objective RL without any assumptions on the coverage of the dataset. As a by-product, our method can be viewed as a highly generalized one as it can easily reduce to certain simpler cases such as CMDPs, which is discussed in Appendix C.

## 1.1 Related works

**Constrained RL.** Our work is closely related to CMDPs (Altman, 1999). Several recent works (Efroni et al., 2020; Ding et al., 2021; Qiu et al., 2020; Brantley et al., 2020; Tessler et al., 2018) have studied the MDP with linear constraints and others (Bhatnagar and Lakshmanan, 2012; Chow et al., 2017; Paternain et al., 2019) have considered RL with more complex constraints. Most of them have provided algorithms with both regret and total constraint violation guarantees. However, while these approaches are successful in CMDPs, they cannot be applied to CMOMDPs where objectives are multiple and constraints can be nonlinear. The difference between their methods and ours are (i) they usually assume the Slater's condition, while we consider its geometric analog, and (ii) most works utilize the Lagrangian multiplier to handle constraints, while in our method, we obtain a primal-dual formulation via conjugate.

**Multi-objective RL.** Existing studies on multi-objective RL can be divided into two groups: the first maintains a set of Pareto-optimal policies (Barrett and Narayanan, 2008; Castelletti et al., 2011, 2012; Wang and Sebag, 2013), and the second collapses multi-objective vector into a scalar and then apply a standard RL method (Barrett and Narayanan, 2008; Natarajan and Tadepalli, 2005; Van Moffaert et al., 2013; Cheung, 2019). However, these methods are available in unconstrained multi-objective RL but unsuitable for constrained cases. The difficulty is that the constraints invalidate the Bellman optimality principle, which is the cornerstone of these works.

**Offline RL.** Recent successful offline RL (Lange et al., 2012; Levine et al., 2020) methods (Fujimoto et al., 2019; Laroche et al., 2019; Jaques et al., 2019; Wu et al., 2019; Kumar et al., 2019, 2020; Agarwal et al., 2020; Yu et al., 2020; Kidambi et al., 2020; Siegel et al., 2020; Nair et al., 2020; Liu et al., 2020; Wang et al., 2020a,b; Tosatto et al., 2017; Farahmand et al., 2016, 2010) fall into two categories (Buckman et al., 2020): (i) uncertainty-aware pessimistic algorithms and (ii) proximal pessimistic algorithms. (i) gives a lower estimate for states and actions less covered by the dataset, while (ii) avoids visiting these less visited states and actions by regularization. However, existing methods are largely based on approximate dynamic programming and do not take constraints into account, thus failing to apply. Our proposed method generalizes the ordinary pessimistic algorithm so that it is compatible with multiple objectives and constraints.

In the analysis, we only assume the compliance of the dataset. In comparison with existing works, which assume the sufficient coverage of the dataset such as the finite concentrability coefficients (Antos et al., 2008; Farahmand et al., 2010; Scherrer et al., 2015; Le et al., 2019; Chen and Jiang, 2019; Fu et al., 2020) and uniformly lower bounded densities of visitation measures (Yin et al., 2021), we require no assumptions as they often fail to hold in practice. As Wang et al. (2020a) have studied the influence of insufficient coverage of the dataset, their conclusion is also reflected in our main results. Moreover, we do not impose any constraints on the affinity of the behavior policy and the learned policy (Liu et al., 2020). In addition, our general algorithm uses convex conjugate, a technique used similarly in Yu et al. (2021); Miryoosefi and Jin (2021) but their method cannot apply to offline situations and Yu et al. (2021) do not address the function approximation setting.

## 1.2 Notations

For simplicity, we write $\langle f, g \rangle_{\mathcal{X}} = \int_{\mathcal{X}} f(x)g(x)\,\mathrm{d}x$ as the inner product on space $\mathcal{X}$ for functions $f, g : \mathcal{X} \to \mathbb{R}$. We also define the norms $\|f\|_{2,\mathcal{X}} = \sqrt{\langle f, f \rangle_{\mathcal{X}}}$ and $\|f\|_{\infty,\mathcal{X}} = \sup_{x \in \mathcal{X}} |f(x)|$. We denote by $[n]$ the set of positive integers not greater than $n$, i.e., $[n] = \{i \in \mathbb{Z} \,|\, 1 \leq i \leq n\}$. For two vectors $\boldsymbol{x}, \boldsymbol{x}' \in \mathbb{R}^D$, we write $\boldsymbol{x} \geq \boldsymbol{x}'$ (or $\boldsymbol{x} \leq \boldsymbol{x}'$) if $\boldsymbol{x}$ is not smaller (or larger) than $\boldsymbol{x}'$ in an elementwise manner. We set $|\boldsymbol{x}| = (|x_1|, |x_2|, \ldots, |x_D|)^{\top}$ and $\boldsymbol{x}_+ = (\max\{x_1, 0\}, \max\{x_2, 0\}, \ldots, \max\{x_D, 0\})^{\top}$. We denote by $\mathcal{B}^D$ the $D$-dimensional unit Euclidean ball, i.e., $\mathcal{B}^D = \{\boldsymbol{x} \in \mathbb{R}^D : \|\boldsymbol{x}\|_2 \leq 1\}$. The set of probability distributions on a set $\mathcal{X}$ is denoted by $\Delta(\mathcal{X})$. We denote by $\widetilde{O}(\cdot)$ the big $O$ notation ignoring logarithmic factors.

## 2 Preliminaries

### 2.1 Constrained multi-objective Markov decision process (CMOMDP)

We consider an episodic CMOMDP given by $\mathcal{M} = (\mathcal{S}, \mathcal{A}, H, \mathcal{P}, \boldsymbol{c})$ where $\mathcal{S}$ is a state space, $\mathcal{A}$ is an action space, $H \in \mathbb{N}_+$ is the horizon, $\mathcal{P} = \{\mathcal{P}_h\}_{h=1}^H$ is a collection of transition kernels $\mathcal{P}_h : \mathcal{S} \times \mathcal{A} \to \Delta(\mathcal{S})$, and $\boldsymbol{c} = \{\boldsymbol{c}_h\}_{h=1}^H$ is a collection of cost functions $\boldsymbol{c}_h : \mathcal{S} \times \mathcal{A} \to [0, 1]^D$. Note that by $\mathcal{P}$ we mean both the probability density function and the probability mass function, as they can be unified (see Appendix A for details). We suppose a fixed initial state $\underline{s} \in \mathcal{S}$ for simplicity. However, a stochastic initial state poses no extra difficulty to our analysis. In each episode, given a policy $\pi = \{\pi_h\}_{h=1}^H$ where $\pi_h : \mathcal{S} \to \Delta(\mathcal{A})$, the agent interacts with the environment as follows. At state $s_h$, the agent takes action $a_h \in \mathcal{A}$ according to $\pi_h(\cdot \,|\, s_h)$, and then receives a stochastic cost $\boldsymbol{c}_h \in [0, 1]^D$, for which we assume $\mathbb{E}[\boldsymbol{c}_h | s_h, a_h] = \boldsymbol{c}_h(s, a)$. The system then transits to the next state $s_{h+1}$ according to $\mathcal{P}_h(\cdot \,|\, s_h, a_h)$. The episode terminates at state $s_{H+1}$ where no action is taken and the cost is zero.

Given a policy $\pi$, the state-value function $\boldsymbol{V}_h^\pi : \mathcal{S} \to [0, H]^D$ and the action-value function $\boldsymbol{Q}_h^\pi : \mathcal{S} \times \mathcal{A} \to [0, H]^D$ at the $h$-th step are defined as $\boldsymbol{V}_h^\pi(s) = \mathbb{E}_\pi[\sum_{i=h}^H \boldsymbol{c}_i(s_i, a_i) \,|\, s_h = s]$ and $\boldsymbol{Q}_h^\pi(s, a) = \mathbb{E}_\pi[\sum_{i=h}^H \boldsymbol{c}_i(s_i, a_i) \,|\, s_h = s, a_h = a]$ where the expectation $\mathbb{E}_\pi$ is taken with respect to $a_i \sim \pi_i(\cdot \,|\, s_i)$ and $s_{i+1} \sim \mathcal{P}(\cdot \,|\, s_i, a_i)$ for $i \in [h : H]$. Note that those value functions are $D$-dimensional vector-valued, and we use bold letters to represent vectors. We denote their $i$-th scalar components by $V_h^{i,\pi}$ and $Q_h^{i,\pi}$ respectively. For notational simplicity, we write: $\mathbb{P}_h[\boldsymbol{V}](s, a) = \mathbb{E}_{s' \sim \mathcal{P}_h(\cdot \,|\, s, a)}[\boldsymbol{V}(s')]$, $\mathbb{B}_h[\boldsymbol{V}](s, a) = \boldsymbol{c}_h(s, a) + \mathbb{P}_h[\boldsymbol{V}](s, a)$, and $\mathbb{D}_\pi[\boldsymbol{Q}](s) = \mathbb{E}_{a \sim \pi(\cdot \,|\, s)}[\boldsymbol{Q}(s, a)]$. Thereby, the Bellman equation for any $(s, a) \in \mathcal{S} \times \mathcal{A}$ and $h \in [H]$ can be written as $\boldsymbol{Q}_h^\pi(s, a) = \mathbb{B}_h[\boldsymbol{V}_h^\pi](s, a)$, and $\boldsymbol{V}_h^\pi(s) = \mathbb{D}_\pi[\boldsymbol{Q}_h^\pi](s)$. Our goal is to minimize the value of a preference function subject to some constraints. Formally, we define $g : \mathbb{R}^D \to \mathbb{R}$ as the preference function and assume that $g$ is 1-Lipschitz and convex and that $g(\boldsymbol{x}) \geq g(\boldsymbol{x}')$ holds as long as $\boldsymbol{x} \geq \boldsymbol{x}'$. For example, $g(\boldsymbol{x})$ can be the summation over $\{x_i\}_{i=1}^n$ or the maximum among $\{x_i\}_{i=1}^n$. We aim to find the optimal policy $\pi^*$ of the following constrained optimization problem,

$$\min_{\pi \in \Delta(\mathcal{A} \,|\, \mathcal{S}, H)} g\big(\boldsymbol{V}_1^\pi(\underline{s})\big) \quad \text{s.t.} \quad \boldsymbol{V}_1^\pi(\underline{s}) \in \mathcal{W}^*, \tag{1}$$

where $\mathcal{W}^* \subseteq [0, H]^D$ is a target set. We use $\boldsymbol{V}_1^*(\underline{s}) = \boldsymbol{V}_1^{\pi^*}(\underline{s})$ to denote the state-value function of the optimal policy of (1) for simplicity.

Solving (1) is impossible when we have no geometric requirements on the target set $\mathcal{W}^*$. Corresponding to the Slater's condition[1] commonly imposed in CMDPs, we establish its geometric analogue for CMOMDPs, as studied in Yu et al. (2021). Let $\mathcal{V}$ be the set of achievable values, i.e., $\mathcal{V} = \{\boldsymbol{V}_1^\pi(\underline{s}) : \text{any policy } \pi\}$, and $\mathcal{W} = \mathcal{V} \cap \mathcal{W}^*$ be the achievable values within the target set. We suppose $\mathcal{W}$ is nonempty. We denote by $\partial \mathcal{W}^*$ and $\partial \mathcal{V}$ the boundaries of $\mathcal{W}^*$ and $\mathcal{V}$, respectively. With a little abuse of notations, we set $\partial \mathcal{W} = \partial \mathcal{W}^* \cap \partial \mathcal{V}$ as the intersection of boundaries. If $\partial \mathcal{W}$ is nonempty, then for each $\boldsymbol{W} \in \partial \mathcal{W}$, we define the angle between support vectors[2] at $\boldsymbol{W}$ as

$$\gamma(\boldsymbol{W}) = \min \big\{ \angle(\boldsymbol{a}, \boldsymbol{b}) \,|\, \boldsymbol{a}, \boldsymbol{b} \text{ are support vectors of } \mathcal{W}^* \text{ and } \mathcal{V} \text{ at } \boldsymbol{W}, \text{ respectively} \big\}$$

where $\angle(\boldsymbol{a}, \boldsymbol{b})$ denotes the angle between $\boldsymbol{a}$ and $\boldsymbol{b}$. We impose geometric requirements on the target set $\mathcal{W}^*$ as stated below.

**Assumption 1.** *We assume the target set $\mathcal{W}^*$ is closed and convex, and there exists an upper bound $\gamma_{\max} \in [\frac{\pi}{2}, \pi)$ such that $\max_{\boldsymbol{W} \in \partial \mathcal{W}} \gamma(\boldsymbol{W}) < \gamma_{\max}$. Moreover, we assume $\mathcal{W}^*$ is a lower set, i.e., for any $\boldsymbol{W} \in \mathcal{W}^*$, it holds that $\boldsymbol{W}' \in \mathcal{W}^*$ for any $\boldsymbol{0} \leq \boldsymbol{W}' \leq \boldsymbol{W}$.*

We write $\rho = 2/\sin(\gamma_{\max})$ for notational simplicity. Explanations and justifications of Assumption 1 are provided in Appendix B where we show that this assumption is reasonable and closely related to Slater's condition. However, the optimization problem (1) is still hard even with these assumptions. Instead, we solve it by considering the suboptimality and constraint violation as performance metrics, which are defined as

$$\text{SubOpt}(\pi) = g\big(\boldsymbol{V}_1^\pi(\underline{s})\big) - g\big(\boldsymbol{V}_1^*(\underline{s})\big), \quad \text{Violation}(\pi) = \text{dist}\big(\boldsymbol{V}_1^\pi(\underline{s}), \mathcal{W}^*\big), \tag{2}$$

---

[1] For CMDPs with constraints in the form of $\boldsymbol{V}^\pi \leq \boldsymbol{b}$, the Slater's condition assumes the existence of a policy $\pi$ such that $\boldsymbol{V}^\pi < \boldsymbol{b}$.

[2] The support vectors at a point are normals to the support hyperplanes at this point.

where we define $\mathrm{dist}(\boldsymbol{V}_1^\pi(\underline{s}), \mathcal{W}^*) = \min_{\boldsymbol{W} \in \mathcal{W}^*} \|\boldsymbol{V}_1^\pi(\underline{s}) - \boldsymbol{W}\|_2$.

To show that the formulation of CMOMDPs is reasonable, we state the relationship between CMOMDPs and CMDPs in the following proposition, which suggests that the CMOMDP is a generalization of the CMDP. The proof is deterred to Appendix C.1.

**Proposition 1.** *The CMDP is a special case of the CMOMDP, and the Slater's condition assumed in CMDPs corresponds to the existence of $\gamma_{\max}$ in Assumption 1 in CMOMDPs.*

### 2.2 Offline learning

We consider the offline setting where we only have access to a dataset $\mathcal{D} = \{(s_h^\tau, a_h^\tau, \boldsymbol{c}_h^\tau)\}_{h,\tau=1}^{H,N}$ with $N$ trajectories collected a priori by an experimentor. We only make the following assumption on the data collecting process.

**Assumption 2** (Compliance of dataset)**.** *We assume that the dataset $\mathcal{D}$ is compliant with the CMOMDP $\mathcal{M}$, i.e., for any $\boldsymbol{c} \in [0,1]^D, s' \in \mathcal{S}, h \in [H]$, and $\tau \in [N]$,*

$$\mathrm{Pr}_{\mathcal{D}}\big(s_{h+1}^\tau = s', \boldsymbol{c}_h^\tau = \boldsymbol{c} \,\big|\, \mathcal{F}_h^\tau\big) = \mathrm{Pr}\big(s_{h+1} = s', \boldsymbol{c}_h = \boldsymbol{c} \,\big|\, s_h = s_h^\tau, a_h = a_h^\tau\big), \tag{3}$$

*where $\mathcal{F}_h^\tau$ is the $\sigma$-algebra generated via $\mathcal{F}_h^\tau = \sigma(\{(s_i^n, a_i^n, c_i^n) \,|\, (n-1)H+i \leq (\tau-1)H+h\})$. The probability on the left-hand side of* (3) *is with respect to the joint distribution of the data collecting process, and that of the right-hand side is with respect to the underlying CMOMDP.*

Assumption 2 is adapted from Jin et al. (2020b). It implies that the dataset $\mathcal{D}$ is generated from the CMOMDP and possesses the Markov property. Such an assumption holds when the experimenter collects the dataset by interacting with the CMOMDP. In particular, we do not require that the data collecting process well explores the state-action space. In other words, we do not impose any assumption on the coverage of the dataset.

## 3 Pessimistic Dual Iteration (PEDI)

In this section, we first motivate our method to solve CMOMDPs in Section 3.1. Then, we develop our algorithm, Pessimistic Dual Iteration (PEDI), for general CMOMDPs in Section 3.2 and introduce an instantiation for linear kernel CMOMDPs in Section 3.3.

### 3.1 Primal-dual formulation

To solve problems like (1), recent works usually apply Lagrangian multipliers to handle the constraints. However, we seek to use another dual method. Specifically, we consider the following problem:

$$\mathsf{p}^* = \min_\pi \big(\mathrm{SubOpt}(\pi) + \nu \,\mathrm{Violation}(\pi)\big), \tag{4}$$

where $\nu$ is a scaling constant that serves as a trade-off between suboptimality and constraint violation. In Appendix D, we show that when $\nu > 1$, (1) and (4) share the solution, and thus they are equivalent. We note that the formulation of (4) is different from existing works on CMDPs where $\nu$ usually plays the role of the Lagrangian multiplier. Intuitively, a large $\nu$ lies more emphasis on satisfying the constraints, while a small $\nu$ focus on reducing the suboptimality.

However, $\mathsf{p}^*$ is hard to solve in (4), since the relationship between the objective and the policy $\pi$ is complex. Therefore, we substitute the policy $\pi$ with the state-value function in the following way,

$$\begin{aligned}
\mathsf{p}^* &= \min_\pi \Big(g\big(\boldsymbol{V}_1^\pi(\underline{s})\big) - g\big(\boldsymbol{V}_1^*(\underline{s})\big) + \nu \,\mathrm{dist}\big(\boldsymbol{V}_1^\pi(\underline{s}), \mathcal{W}^*\big)\Big) \\
&= \min_{\boldsymbol{V} \in \mathcal{V}} \big(g(\boldsymbol{V}) - g\big(\boldsymbol{V}_1^*(\underline{s})\big) + \nu \,\mathrm{dist}(\boldsymbol{V}, \mathcal{W}^*)\big)
\end{aligned} \tag{5}$$

Nevertheless, solving this problem is still hard since the objective is nonlinear in $\boldsymbol{V}$ and thus standard RL cannot apply. Therefore, we utilize the convex conjugate to "linearize" the objective. This technique is widely used in related works (Miryoosefi and Jin, 2021; Yu et al., 2021). Specifically, by convex conjugate, we have

$$g(\boldsymbol{V}) = \max_{\boldsymbol{\beta} \in \mathcal{B}^D} \big(\boldsymbol{\beta}^\top \boldsymbol{V} - g^*(\boldsymbol{\beta})\big), \quad \mathrm{dist}(\boldsymbol{V}, \mathcal{W}^*) = \max_{\boldsymbol{\alpha} \in \mathcal{B}^D} \big(\boldsymbol{\alpha}^\top \boldsymbol{V} - \max_{\boldsymbol{x} \in \mathcal{W}^*} \boldsymbol{\alpha}^\top \boldsymbol{x}\big) \tag{6}$$

for any $\boldsymbol{V} \in \mathcal{V}$. The effective domains, $\mathcal{B}^D$, of both $\boldsymbol{\alpha}$ and $\boldsymbol{\beta}$ are obtained by the 1-Lipschitzness and Corollary 13.3.3 in Rockafellar (1970). To see why the second equality of (6) holds, we notice that

$$\text{dist}(\boldsymbol{V}, \mathcal{W}^*) = \max_{\boldsymbol{\alpha} \in \mathcal{B}^D} \left( \boldsymbol{\alpha}^\top \boldsymbol{V} - \underbrace{\max_{\boldsymbol{x} \in \mathbb{R}^D} \left( \boldsymbol{\alpha}^\top \boldsymbol{x} - \text{dist}(\boldsymbol{x}, \mathcal{W}^*) \right)}_{(*)} \right),$$

and $(*)$ attains the maximum when $\boldsymbol{x} \in \mathcal{W}^*$ since $\boldsymbol{\alpha} \in \mathcal{B}^D$. We call $\boldsymbol{\alpha}$ and $\boldsymbol{\beta}$ the dual variables throughout this paper. By plugging (6) into (5), we have

$$\mathsf{p}^* = \min_{\boldsymbol{V} \in \mathcal{V}} \max_{\boldsymbol{\alpha}, \boldsymbol{\beta} \in \mathcal{B}^D} \mathcal{L}(\boldsymbol{V}; \boldsymbol{\alpha}, \boldsymbol{\beta}) = \boldsymbol{\beta}^\top \boldsymbol{V} - g^*(\boldsymbol{\beta}) - g\left(\boldsymbol{V}_1^*(\underline{s})\right) + \nu \boldsymbol{\alpha}^\top \boldsymbol{V} - \nu \max_{\boldsymbol{x} \in \mathcal{W}^*} \boldsymbol{\alpha}^\top \boldsymbol{x}. \tag{7}$$

Its dual problem can be written as $\mathsf{d}^* = \max_{\boldsymbol{\alpha}, \boldsymbol{\beta} \in \mathcal{B}^D} \mathsf{D}(\boldsymbol{\alpha}, \boldsymbol{\beta})$, where $\mathsf{D}(\boldsymbol{\alpha}, \boldsymbol{\beta})$ is the dual function defined as $\mathsf{D}(\boldsymbol{\alpha}, \boldsymbol{\beta}) = \min_{\boldsymbol{V} \in \mathcal{V}} \mathcal{L}(\boldsymbol{V}; \boldsymbol{\alpha}, \boldsymbol{\beta})$. We notice that $\mathcal{L}(\boldsymbol{V}; \boldsymbol{\alpha}, \boldsymbol{\beta})$ is convex in $\boldsymbol{V} \in \mathcal{V}$ and concave in $\boldsymbol{\alpha}, \boldsymbol{\beta} \in \mathcal{B}^D$, which implies the strong duality, $\mathsf{p}^* = \mathsf{d}^*$. Hence, dual methods can apply.

When $\boldsymbol{\alpha}$ and $\boldsymbol{\beta}$ are fixed, solving $\mathsf{D}(\boldsymbol{\alpha}, \boldsymbol{\beta})$ is a planning problem. When the model is known, it suffices to conduct value iteration on the model. However, what we have is a dataset collected a priori in offline RL. Therefore, we develop the offline planning algorithm that is a pessimistic variant of the value iteration (Jin et al., 2020b), which applies penalties to ensure pessimism. We explain why pessimism is crucial to offline CMOMDP in Appendix F. We now describe the high-level intuition of our approach. Consider a meta-algorithm that constructs the empirical transition kernel $\widehat{\mathcal{P}}$ and the empirical cost function $\widehat{c}$ based on the dataset $\mathcal{D}$. As defined below, we introduce the uncertainty quantifier that bounds the estimation error from above with a confidence parameter $\xi \in (0, 1)$.

**Definition 1** ($\xi$-uncertainty quantifier)**.** *We call $\{(\Gamma_h^{\mathcal{P}}, \Gamma_h^{\boldsymbol{c}})\}_{h=1}^H$ a $\xi$-uncertainty quantifier if the following event*

$$\mathcal{E} = \left\{ \left|\widehat{\mathcal{P}}_h(s'|s, a) - \mathcal{P}_h(s'|s, a)\right| \leq \Gamma_h^{\mathcal{P}}(s, a, s'), \ \left|\widehat{c}_h^i(s, a) - c_h^i(s, a)\right| \leq \Gamma_h^{c^i}(s, a) \right.$$
$$\left. \textit{for all } (s, a, s') \in \mathcal{S} \times \mathcal{A} \times \mathcal{S}, h \in [H], i \in [D] \right\} \tag{8}$$

*holds with probability at least $1 - \xi$. Here we write $\Gamma_h^{\boldsymbol{c}} = (\Gamma_h^{c^1}, \Gamma_h^{c^2}, \dots, \Gamma_h^{c^D})$.*

Then, we can construct the empirical counterparts of $\mathbb{P}_h$ and $\mathbb{B}_h$ by $\widehat{\mathbb{P}}_h[\boldsymbol{V}](s, a) = \langle \boldsymbol{V}(\cdot), \widehat{\mathcal{P}}_h(\cdot \mid s, a) \rangle_{\mathcal{S}}$ and $\widehat{\mathbb{B}}_h[\boldsymbol{V}](s, a) = \widehat{\boldsymbol{c}}_h(s, a) + \widehat{\mathbb{P}}_h[\boldsymbol{V}](s, a)$. Note that for any function $V : \mathcal{S} \to [0, H]$, under the event $\mathcal{E}$, it holds for any $(s, a) \in \mathcal{S} \times \mathcal{A}$ and any $h \in [H]$ that

$$\left|(\widehat{\mathbb{P}}_h - \mathbb{P}_h)[V](s, a)\right| = \left| \int_{\mathcal{S}} \left(\widehat{\mathcal{P}}_h(s'|s, a) - \mathcal{P}_h(s'|s, a)\right) V(s') \, \mathrm{d}s' \right| \leq H \int_{\mathcal{S}} \Gamma_h^{\mathcal{P}}(s, a, s') \, \mathrm{d}s'.$$

Therefore, we define $\Gamma_h(s, a) = H \int_{\mathcal{S}} \Gamma_h^{\mathcal{P}}(s, a, s') \, \mathrm{d}s'$, which quantifies the uncertainty arising from the transition kernel. We use the $\xi$-uncertainty quantifier as the penalty function for pessimistic planning, which leads to a conservative estimation of the value function. We formally describe our planning method in Algorithm 1, where $\boldsymbol{\theta}$ denotes a projection vector to be specified later. By adding the pessimistic penalty $\Gamma_h \cdot \mathbf{1} + \Gamma_h^{\boldsymbol{c}}$ to the estimated value, we ensure a conservative estimation. We also truncate $\boldsymbol{Q}_h^k$ and $\boldsymbol{V}_h^k$ in $[0, H - h + 1]^D$ to improve the estimation.

It remains to solve $\max_{\boldsymbol{\alpha}, \boldsymbol{\beta} \in \mathcal{B}^D} \mathsf{D}(\boldsymbol{\alpha}, \boldsymbol{\beta})$. We utilize the projected subgradient method which produces a sequence of dual variables $\{\boldsymbol{\alpha}^k\}_{k=1}^K$ and $\{\boldsymbol{\beta}^k\}_{k=1}^K$ that approximately solves the dual problem. A detailed description of projected subgradient method is provided in Appendix H.6.2. For a brief description, we update the dual variables via

$$\boldsymbol{\alpha}^{k+1} \leftarrow \Pi_{\mathcal{B}^D} \left\{ \boldsymbol{\alpha}^k + \eta^k (\boldsymbol{V}^k - \arg\max_{\boldsymbol{x} \in \mathcal{W}^*} (\boldsymbol{\alpha}^k)^\top \boldsymbol{x}) \right\},$$
$$\boldsymbol{\beta}^{k+1} \leftarrow \Pi_{\mathcal{B}^D} \left\{ \boldsymbol{\beta}^k + \eta^k (\boldsymbol{V}^k - \partial g^*(\boldsymbol{\beta}^k)) \right\}. \tag{9}$$

where $\boldsymbol{V}^k \in \arg\min_{\boldsymbol{V} \in \mathcal{V}} \mathcal{L}(\boldsymbol{V}; \boldsymbol{\alpha}^k, \boldsymbol{\beta}^k)$ and $\eta^k$ is the step length at the $k$-th iteration.

As claimed in Proposition 1, the CMDP is a special case of the CMOMDP. Moreover, we show in Appendix C.2 that when reducing to the CMDP, the objective (7) reduces to the Lagrangian formulation of the CMDP problem. Hence, our proposed method has good generalization ability.

---
**Algorithm 1** Pessimistic planning.
---
1: **function** PESSPLANNING$(\boldsymbol{\theta}, \mathcal{D}, \{(\Gamma_h^{\mathcal{P}}, \Gamma_h^{\boldsymbol{c}})\}_{h=1}^H)$
2:      **for** step $h = H, H-1, \ldots, 1$ **do**
3:          $\overline{\boldsymbol{Q}}_h(\cdot, \cdot) \leftarrow \widehat{\boldsymbol{c}}_h(\cdot, \cdot) + \widehat{\mathbb{P}}_h[\boldsymbol{V}_{h+1}](\cdot, \cdot) + \Gamma_h \cdot \boldsymbol{1} + \Gamma_h^{\boldsymbol{c}}$
4:          $\boldsymbol{Q}_h(\cdot, \cdot) \leftarrow \min\{\overline{\boldsymbol{Q}}_h(\cdot, \cdot), (H-h+1) \cdot \boldsymbol{1}\}_+$
5:          $\pi_h(\cdot) \leftarrow \arg\min_{a \in \mathcal{A}} \boldsymbol{Q}_h(\cdot, \cdot) \cdot \boldsymbol{\theta}$
6:          $\boldsymbol{V}_h(\cdot) \leftarrow \mathbb{D}_{\pi_h}[\boldsymbol{Q}_h](\cdot)$
7:      **end for**
8:      **return** $\pi = \{\pi_h\}_{h=1}^H, \boldsymbol{Q} = \{\boldsymbol{Q}_h\}_{h=1}^H, \boldsymbol{V} = \{\boldsymbol{V}_h\}_{h=1}^H$
9: **end function**
---

## 3.2 General CMOMDPs

Based on the above analysis, we now develop the Pessimistic Dual Iteration (PEDI) (Algorithm 2) for general CMOMDPs, which is motivated by the dual gradient method in solving the minimax optimization problem. Specifically, PEDI solves $D(\boldsymbol{\alpha}, \boldsymbol{\beta})$ via PESSPLANNING (Algorithm 1) and updates dual variables via (9), alternately. The output policy $\widehat{\pi}$ is a mixed policy executed by randomly selecting a policy $\pi^k$ from $\{\pi^k\}_{k=1}^K$ with equal probability beforehand and then exclusively following $\pi^k$ thereafter. Hence, it holds that, for the mixed policy $\widehat{\pi}$, $\boldsymbol{V}_1^{\widehat{\pi}}(\underline{s}) = \frac{1}{K} \sum_{k=1}^K \boldsymbol{V}_1^{\pi^k}(\underline{s})$. Note that the mixed policy may not be a Markov policy (Altman, 1999). However, it would be useful to extend the definition of a policy $\pi = \{\pi_h\}_{h=1}^H$ so as to allow $\pi_h$ to depend not only on $h$ but also on some initial randomizing mechanism. That is, we may have a set of policies $\mathcal{U}$ and a distribution $q \in \Delta(\mathcal{U})$. Then we define the mixed policy of $\mathcal{U}$, $\pi_{\mathcal{U}}$, as a policy to be executed by first using $q$ to choose some policy $\pi \in \mathcal{U}$ and then proceeding with only that policy. The optima of (1) over the mixed policy is still attained by the Markov policy $\pi^*$. Thus, $\mathrm{SubOpt}(\widehat{\pi})$ remains a reasonable performance metric of our algorithm.

---
**Algorithm 2** Pessimistic Dual Iteration (PEDI).
---
**Input:** offline dataset $\mathcal{D}$, step length $\{\eta^k\}_{k=1}^K$, scaling parameter $\nu$
**Output:** the mixed policy $\widehat{\pi} = \{\widehat{\pi}_h\}_{h=1}^H$ of $\{\pi^k\}_{k=1}^K$

1: randomly initialize dual variables $\boldsymbol{\alpha}^1, \boldsymbol{\beta}^1 \in \mathcal{B}^D$, and set $\boldsymbol{\theta}^1 = \nu\boldsymbol{\alpha}^1 + \boldsymbol{\beta}^1$
2: construct $\{\widehat{\boldsymbol{c}}_h\}_{h=1}^H$ and $\{\widehat{\mathcal{P}}_h\}_{h=1}^H$ based on $\mathcal{D}$
3: construct the $\xi$-uncertainty quantifier $\{(\Gamma_h^{\mathcal{P}}, \Gamma_h^{\boldsymbol{c}})\}_{h=1}^H$ based on $\mathcal{D}$
4: **for** iteration $k = 1, 2, \ldots, K$ **do**
5:      $\pi^k, \boldsymbol{Q}^k, \boldsymbol{V}^k \leftarrow$ PESSPLANNING$\left(\boldsymbol{\theta}^k, \mathcal{D}, \{(\Gamma_h^{\mathcal{P}}, \Gamma_h^{\boldsymbol{c}})\}_{h=1}^H\right)$
6:      update $\boldsymbol{\alpha}^k, \boldsymbol{\beta}^k$ to $\boldsymbol{\alpha}^{k+1}, \boldsymbol{\beta}^{k+1}$ via (9)
7:      $\boldsymbol{\theta}^{k+1} \leftarrow \nu\boldsymbol{\alpha}^{k+1} + \boldsymbol{\beta}^{k+1}$
8: **end for**
---

## 3.3 Linear kernel CMOMDPs

In this section, we study PEDI on settings with linear function approximation such as linear kernel CMOMDPs. To that end, we first introduce the linear kernel CMOMDP, which is a generalization of the linear kernel MDP (Yang and Wang, 2019; Jin et al., 2020a; Cai et al., 2020). Then, we propose an instantiation of Algorithm 2 for linear kernel CMOMDPs.

**Definition 2** (Linear kernel CMOMDP). *The CMOMDP $(\mathcal{S}, \mathcal{A}, H, \mathcal{P}, \boldsymbol{c})$ is a linear kernel CMOMDP with a known kernel feature map $\psi : \mathcal{S} \times \mathcal{A} \times \mathcal{S} \to \mathbb{R}^{d_1}$ and a known value feature map $\varphi : \mathcal{S} \times \mathcal{A} \to \mathbb{R}^{d_2}$, if for any $h \in [H]$ there exist unknown vectors $\theta_h \in \mathbb{R}^{d_1}$ and $\theta_h^{c^i} \in \mathbb{R}^{d_2}$ for any $i \in [D]$ such that for any $(s, a, s') \in \mathcal{S} \times \mathcal{A} \times \mathcal{S}$,*

$$\mathcal{P}_h(s' \mid s, a) = \psi(s, a, s')^\top \theta_h, \quad c_h^i(s, a) = \varphi(s, a)^\top \theta_h^{c^i}.$$

*Let $d = \max\{d_1, d_2\}$. We assume there exists a constant $R > 0$ such that*

$$R^{-2} \max_{s' \in \mathcal{S}} \left( y^\top \psi(s, a, s') \right)^2 \le \int_{\mathcal{S}} \left( y^\top \psi(s, a, s') \right)^2 \, \mathrm{d}s' \le d$$

*for any $(s, a) \in \mathcal{S} \times \mathcal{A}$ and $y : \|y\|_2 \le 1$. Moreover, we assume $\|\theta_h\|_2 \le \sqrt{d}$ and $\|\theta_h^{c^i}\|_2 \le \sqrt{d}$ for $i \in [d]$.*

The existence of constant $R$ naturally holds when $\psi(s, a, \cdot)$ is upper bounded and Lipschitz continuous for all $(s, a) \in \mathcal{S} \times \mathcal{A}$. Note that the tabular CMOMDP is a special case of the linear kernel CMOMDP with $R = 1$ (see Appendix G for a detailed discussion).

An instantiation of PEDI for linear kernel CMOMDPs is deferred to Appendix E where we propose a method to estimate the empirical transition kernel $\widehat{\mathcal{P}}$ and cost function $\widehat{c}$ and thereby construct a $\xi$-uncertainty quantifier.

## 4 Theoretical results

In this section, we first show upper bounds for suboptimality and constraint violation for PEDI on general CMOMDPs in Section 4.1. In Section 4.2, we show that PEDI achieves the minimax optimality up to the optimization error and multiplicative factors of constants related to the CMOMDP when specified to linear kernel CMOMDPs. Moreover, when the dataset has sufficient coverage over the trajectories induced by the optimal policy, the intrinsic uncertainty is guaranteed to be $\widetilde{\mathcal{O}}(1/\sqrt{N})$ where $N$ is the number of trajectories in the dataset.

### 4.1 Results of general CMOMDPs

By using pessimistic planning (Algorithm 1), we ensure a pessimistic policy $\widehat{\pi}$. We formally state this notion in the following lemma.

**Lemma 1** (Pessimism)**.** *Suppose that Assumptions 1 and 2 hold. By Algorithm 2, under event $\mathcal{E}$ defined in (8), for any $(s, a) \in \mathcal{S} \times \mathcal{A}$ and $h \in [H]$, it holds that $\boldsymbol{V}_h^k(s) \ge \boldsymbol{V}_h^{\pi^k}(s)$, $\boldsymbol{Q}_h^k(s, a) \ge \boldsymbol{Q}_h^{\pi^k}(s, a)$, and $\mathrm{dist}\left(\boldsymbol{V}_1^k(\underline{s}), \mathcal{W}^*\right) \ge \mathrm{dist}\left(\boldsymbol{V}_1^{\pi^k}(\underline{s}), \mathcal{W}^*\right)$. Furthermore, it holds that*

$$g\left(\widehat{\boldsymbol{V}}_1(\underline{s})\right) \ge g\left(\boldsymbol{V}_1^{\widehat{\pi}}(\underline{s})\right) \quad \text{and} \quad \mathrm{dist}\left(\widehat{\boldsymbol{V}}_1(\underline{s}), \mathcal{W}^*\right) \ge \mathrm{dist}\left(\boldsymbol{V}_1^*(\underline{s}), \mathcal{W}^*\right),$$

*where $\widehat{\boldsymbol{V}}_1(\underline{s}) = \frac{1}{K} \sum_{k=1}^K \boldsymbol{V}_1^k(\underline{s})$ and $\widehat{\pi}$ is the output of Algorithm 2.*

*Proof of Lemma 1.* See Appendix H.1 for a detailed proof. $\qquad\square$

We are now ready to establish the following theorem, which characterizes the suboptimality and constraint violation of PEDI for general offline CMOMDPs.

**Theorem 1** (Suboptimality and constraint violation for general CMOMDPs)**.** *Suppose that Assumptions 1 and 2 hold. Under event $\mathcal{E}$ defined in (8), for the output policy $\widehat{\pi}$ of Algorithm 2, when we set $\nu = \rho$ and $\eta^k = 2G^{-1}\sqrt{D/k}$ (or $2G^{-1}\sqrt{D/K}$ if $K$ is predefined), where $G = 2(1+\nu)H\sqrt{D}$, it holds that*

$$\mathrm{SubOpt}(\widehat{\pi}) \le \epsilon_K + \mathrm{IntUncert}_{\mathcal{D}}^{\pi^*}, \quad \mathrm{Violation}(\widehat{\pi}) \le \frac{2}{\rho}(\epsilon_K + \mathrm{IntUncert}_{\mathcal{D}}^{\pi^*}), \tag{10}$$

*where we let $C$ denote an absolute constant and define*

$$\epsilon_K = C(1+\rho)\sqrt{\frac{DH^2}{K}}, \quad \mathrm{IntUncert}_{\mathcal{D}}^{\pi^*} = 2(1+\rho)\sqrt{D} \sum_{h=1}^H \mathbb{E}_{\pi^*}\left[\Gamma_h(s_h, a_h) + \|\Gamma_h^c(s_h, a_h)\|_\infty \mid s_1 = \underline{s}\right].$$

*Proof of Theorem 1.* See Appendix H.2 for a detailed proof. Here we provide a proof sketch, which consists of two steps.

**First step**: upper bounding $\mathrm{SubOpt}(\widehat{\pi}) + \nu \, \mathrm{Violation}(\widehat{\pi})$, which is our objective (4). Here $\widehat{\pi}$ is the output of PEDI (Algorithm 2). By pessimism, we have $\mathrm{SubOpt}(\widehat{\pi}) + \nu \, \mathrm{Violation}(\widehat{\pi}) \le$

$g(\widehat{\boldsymbol{V}}_1(\underline{s})) - g(\boldsymbol{V}_1^*(\underline{s})) + \nu \operatorname{dist}(\widehat{\boldsymbol{V}}_1(\underline{s}), \mathcal{W}^*)$ where $\widehat{\boldsymbol{V}}_1(\underline{s}) = \frac{1}{K} \sum_{k=1}^K \boldsymbol{V}_1^k(\underline{s})$. We do linearization via convex conjugate (6) and then get

$$\operatorname{SubOpt}(\hat{\pi}) + \nu \operatorname{Violation}(\hat{\pi}) \leq K^{-1} \max_{\|\boldsymbol{\beta}\| \leq 1} \left\{ \boldsymbol{\beta} \cdot \sum_{k=1}^K \boldsymbol{V}_1^k(\underline{s}) - \sum_{k=1}^K g^*(\boldsymbol{\beta}) \right\} - g(\boldsymbol{V}_1^*(\underline{s}))$$
$$+ K^{-1}\nu \max_{\|\boldsymbol{\alpha}\| \leq 1} \left\{ \boldsymbol{\alpha} \cdot \sum_{k=1}^K \boldsymbol{V}_1^k(\underline{s}) - \sum_{k=1}^K \max_{\boldsymbol{x} \in \mathcal{W}^*} \boldsymbol{\alpha} \cdot \boldsymbol{x} \right\}.$$

Recall that PEDI uses the projected subgradient method, which approximates the max operator above by a sequence of variables at the expense of a sublinear approximation error, which leads to

$$\operatorname{SubOpt}(\hat{\pi}) + \nu \operatorname{Violation}(\hat{\pi}) \leq K^{-1} \sum_{k=1}^K \left\{ \left(\boldsymbol{\beta}^k\right)^\top \boldsymbol{V}_1^k(\underline{s}) - g^*\left(\boldsymbol{\beta}^k\right) \right\} - g(\boldsymbol{V}_1^*(\underline{s}))$$
$$+ K^{-1}\nu \sum_{k=1}^K \left\{ \left(\boldsymbol{\alpha}^k\right)^\top \boldsymbol{V}_1^k(\underline{s}) - \max_{\boldsymbol{x} \in \mathcal{W}^*} \left(\boldsymbol{\alpha}^k\right)^\top \boldsymbol{x} \right\}$$
$$+ C(1+\nu)\sqrt{DH^2/K},$$

where $C$ is a constant. We rearrange the right-hand side above with some tricks and then obtain

$$\operatorname{SubOpt}(\hat{\pi}) + \nu \operatorname{Violation}(\hat{\pi}) \leq K^{-1} \sum_{k=1}^K \left[ \left(\boldsymbol{\theta}^k\right)^\top \left(\boldsymbol{V}_1^k(\underline{s}) - \boldsymbol{V}_1^*(\underline{s})\right) \right] + C(1+\nu)\sqrt{DH^2/K}.$$

The last term in the right hand side above is exactly $\epsilon_K$. Intuitively, the term $(\boldsymbol{\theta}^k)^\top (\boldsymbol{V}_1^k(\underline{s}) - \boldsymbol{V}_1^*(\underline{s}))$ indicates the projected difference of state-value functions along $\boldsymbol{\theta}$, which motivates the PESSPLANNING (Algorithm 1). We apply results from this pessimistic approach and then conclude

$$\operatorname{SubOpt}(\hat{\pi}) + \nu \operatorname{Violation}(\hat{\pi}) = g\left(\boldsymbol{V}_1^{\hat{\pi}}(\underline{s})\right) - g\left(\boldsymbol{V}_1^*(\underline{s})\right) + \nu \operatorname{dist}\left(\boldsymbol{V}_1^{\hat{\pi}}(\underline{s}), \mathcal{W}^*\right) \leq \epsilon_K + \operatorname{IntUncert}_{\mathcal{D}}^{\pi^*}.$$

**Second step**: upper bounding $\operatorname{SubOpt}(\hat{\pi})$ and $\operatorname{Violation}(\hat{\pi})$ individually. This is done by two facts: (i) $\operatorname{dist}(\cdot, \cdot) \geq 0$, and (ii) $g(\boldsymbol{V}_1^{\hat{\pi}}(\underline{s})) - g(\boldsymbol{V}_1^*(\underline{s})) \geq -\nu \operatorname{dist}(\boldsymbol{V}_1^{\hat{\pi}}(\underline{s}), \boldsymbol{V}_1^*(\underline{s}))/2$. We plug them into the resulting inequality in the first step and then complete the proof. □

As shown in Theorem 1, our pessimistic algorithm bounds the suboptimality and constraint violation from above by the sum of the optimization error $\epsilon_K \in \mathcal{O}(1/\sqrt{K})$ and the intrinsic uncertainty $\operatorname{IntUncert}_{\mathcal{D}}^{\pi^*}$. The optimization error $\epsilon_K$ vanishes as $K$ goes into infinity. Later, we will show that the intrinsic uncertainty, $\operatorname{IntUncert}_{\mathcal{D}}^{\pi^*}$, is impossible to eliminate through an analysis of PEDI on linear kernel CMOMDPs in Section 4.2.

### 4.2 Results of linear kernel CMOMDPs

The following theorem characterizes the suboptimality and constraint violation of PEDI for linear kernel CMOMDPs. Recall that we instantiate PEDI to linear kernel CMOMDPs in Appendix E.

**Theorem 2** (Suboptimality and constraint violation for linear kernel CMOMDPs). *We set $\lambda = 1$ and $\kappa = CR\sqrt{d \log(dN) + \log(DH/\xi)}$ where $C > 0$ is an absolute constant. Then, under Assumptions 1 and 2, $\{\Gamma_h^{\mathcal{P}}, \Gamma_h^r\}_{h=1}^H$ defined in (24) (Appendix E) is a $\xi$-uncertainty quantifier. Hence, under the same settings of $\nu$ and $\eta^k$ in Theorem 1, (10) holds with probability at least $1 - \xi$ for the output $\hat{\pi}$, and the intrinsic uncertainty is specified to*

$$\operatorname{IntUncert}_{\mathcal{D}}^{\pi^*} = 2(1+\rho)\sqrt{D} \sum_{h=1}^H \mathbb{E}_{\pi^*}\left[ H \int_{\mathcal{S}} \kappa \cdot \|\psi(s_h, a_h, s')\|_{\Lambda_h^{-1}} \mathrm{d}s' + \kappa \cdot \|\varphi(s_h, a_h)\|_{\Lambda_{\varphi,h}^{-1}} \Big| s_1 = \underline{s} \right].$$

*Proof of Theorem 2.* See Appendix H.3 for a detailed proof. □

In what follows, we show that when the dataset sufficiently covers the trajectories of the optimal policy, the intrinsic uncertainty is $\mathcal{O}(1/\sqrt{N})$ where $N$ is the number of trajectories in the dataset.

**Corollary 1** (Dataset with sufficient coverage). *Suppose that the $\tau$-th trajectory $\left\{(s_h^\tau, a_h^\tau, \boldsymbol{c}_h^\tau)\right\}_{h=1}^{H}$ of the dataset $\mathcal{D}$ is generated by a behavior policy $\pi^{\mathrm{b},\tau}$ for $\tau \in [N]$. Meanwhile, we assume that there exists a constant $\varsigma > 0$ such that $\mu_h^*(s,a)/\mu_h^{\mathrm{b},\tau}(s,a) \leq \varsigma$ for any $h \in [H], \tau \in [N]$, and $(s,a) \in \mathcal{S} \times \mathcal{A}$. Here $\{\mu_h^*\}_{h=1}^{H}$ and $\{\mu_h^{\mathrm{b},\tau}\}_{h=1}^{H}$ are the visitation measures of $\pi^*$ and $\pi^{\mathrm{b},\tau}$. Then, we have with probability at least $1 - \xi$ that*

$$\mathrm{IntUncert}_{\mathcal{D}}^{\pi^*} \in \widetilde{\mathcal{O}}\big(C\varsigma\kappa(1+\rho)H^2\sqrt{d|\mathcal{S}|/N}\big).$$

*Proof of Corollary 1.* See Appendix H.5 for a detailed proof. □

Here $|\mathcal{S}|$ denotes a certain measure on $\mathcal{S}$. For instance, when $\mathcal{S} = [0,1]$ and we take the Lebesgue measure, we have $|\mathcal{S}| = 1$. Note that Corollary 1 holds under a weaker condition than the uniform coverage (Wang et al., 2020a; Rashidinejad et al., 2021), where the condition is $\max_\pi \mu^\pi/\mu^{\mathrm{b}} \leq C$ for a certain constant $C$. However, our algorithm is still able to achieve the intrinsic uncertainty of $\widetilde{O}(\sqrt{1/N})$. In particular, our condition, $\mu^*/\mu^{\mathrm{b}} \leq \varsigma$, can hold when we have expert demonstration in the dataset $\mathcal{D}$ (Buckman et al., 2020).

In Theorem 2, we notice that when the number of iteration $K$ goes into infinity, the optimization error $\epsilon_K$ vanishes, and the suboptimality and constraint violation are bounded from above by the intrinsic uncertainty $\mathrm{IntUncert}_{\mathcal{D}}^{\pi^*}$ only. However, this is the best effort that we can put. The following theorem indicates that the term $\mathrm{IntUncert}_{\mathcal{D}}^{\pi^*}$ is impossible to eliminate since it arises from the information-theoretic lower bound.

**Theorem 3** (Information-theoretic lower bound). *For the output $\widehat{\pi}$ of any algorithm of offline CMOMDPs, there exists a linear kernel CMOMDP $\mathcal{M}$ with initial state $\underline{s} \in \mathcal{S}$ and a dataset $\mathcal{D}$ compliant with $\mathcal{M}$, such that*

$$\mathbb{E}_{\mathcal{D}}\left[\frac{\mathrm{SubOpt}(\widehat{\pi})}{\sum_{h=1}^{H}\mathbb{E}_{\pi^*}\left[\|\varphi(s_h,a_h)\|_{\Lambda_{\varphi,h}^{-1}} + |\mathcal{S}|^{-1}\int_{\mathcal{S}}\|\psi(s_h,a_h,s')\|_{\Lambda_h^{-1}}\,\mathrm{d}s' \,\Big|\, s_1 = \underline{s}\right]}\right] \geq c,$$

*where the expectation is taken with respect to the data collecting process, and $c > 0$ is an absolute constant.*

*Proof of Theorem 3.* See Appendix H.4 for a detailed proof. □

By Theorem 2 and Theorem 3, we find that the upper bound of suboptimality of Algorithm 2 for linear kernel CMOMDP matches the information-theoretic lower bound up to the optimization error $\mathcal{O}(1/\sqrt{K})$ and some multiplicative factors of horizons, dimensions, and geometric properties of the target set, which means that PEDI is near-optimal for linear kernel CMOMDP.

## 5 Experimental Results

We also conduct numerical experiments to verify our theory. Please see Appendix I for details. The results show that the proposed algorithm is not only provably efficient but also applicable.

## 6 Conclusion

In this paper, we propose a general algorithm, Pessimistic Dual Iteration (PEDI), for constrained multi-objective reinforcement learning with offline data. We show our algorithm delivers a data-dependent upper bound for suboptimality and constraint violation. The upper bound is composed of the optimization error and the intrinsic uncertainty. Moreover, we show that the bound of suboptimality is minimax optimal for linear kernel CMOMDPs up to the optimization error and multiplicative factors of constants related to the CMOMDP. When the trajectories of optimal policy are sufficiently covered, the intrinsic uncertainty is proved to be $\widetilde{\mathcal{O}}(1/\sqrt{N})$ where $N$ is the number of trajectories in the dataset.

## Acknowledgments and Disclosure of Funding

We thank all anonymous reviewers for their useful comments. Zhaoran Wang acknowledges National Science Foundation (Awards 2048075, 2008827, 2015568, 1934931), Simons Institute (Theory of Reinforcement Learning), Amazon, J.P. Morgan, and Two Sigma for their supports. Zhuoran Yang acknowledges Simons Institute (Theory of Reinforcement Learning).

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
