# OpenReview forum: "Offline Constrained Multi-Objective Reinforcement Learning via Pessimistic Dual Value Iteration"
_NeurIPS.cc/2021/Conference — NeurIPS 2021 Poster_

### Official Review · Reviewer_U6xP · 2021-06-27

**Rating:** 6
**Confidence:** 3

**Summary:**

The paper considers policy optimization in a difficult setting involving multiple challenges: (1) multi-objective -- rather than simply optimizing a scalar reward, the policy must optimize a convex function g on vectors of cumulative costs; (2) constrained -- while optimizing g the policy must also satisfy a constraint on the cumulative costs; (3) offline -- the policy learning algorithm cannot actively query the environment, and only has offline access to experience.

For this setting, the paper presents an algorithm with corresponding theoretical analysis. The algorithm follows a primal-dual optimization to handle (1) and (2) above while proposing pessimistic value iteration to handle (3) above. Theoretical analysis provides guarantees on the convergence of the algorithm to an optimal policy under idealized conditions in both general MDPs and kernelized MDPs.

**Main Review:**

The paper is written well, especially for a theoretical paper. I appreciate the time the authors took to organize and polish the writing to this point. Content-wise, the theoretical analysis appears detailed, clean, and comprehensive.

The main issue with the current paper is significance, which I elaborate on below, considering three dimensions:

(A) Algorithmically: Algorithmically, the proposed techniques are only marginally novel. Certainly, pessimistic approaches to offline RL are ubiquitous, as detailed in the paper's related work, and these existing approaches are arguably identical to the paper's proposed approach. Similarly, primal-dual approaches have been proposed for constrained RL, although I concede that, as the paper notes, these previous works typically consider linear constraints/cost functions.

(B) Theoretically: The contributions theoretically appear to be (1) formalization of the CMOMDP problem and (2) guarantees on the proposed algorithm. These contributions largely appear to be straightforward extensions of previous work (e.g., the CMOMDP is essentially a combination of C(onstrained)MDPs and M(ulti)O(bjective)MDPs). It is unclear if there are any insights that come out from these generalizations of existing guarantees, beyond the ideas behind Assumption 1 (generalization of Slater's conditions), which I concede is a significant & novel contribution.

(C) Experimentally: The paper does not include any empirical analysis of the algorithm. Personally, as someone who has practical experience with multi-objective, constrained, and offline RL algorithms, I know how difficult it is to get such algorithms to work empirically. An algorithm as complex as the one proposed in the paper is further unlikely to work well in practice, so without empirical evaluations, the reader is left to assume that the algorithm is exclusively theoretical, with no practical ramifications.

**Time Spent Reviewing:**

3

---

> ### Author Response · Authors · 2021-08-10
> **Response**
>
> Thank you for acknowledging the strong results of this work and the quality of the paper. We address your comments as follows in detail.
>
> > Q1: Certainly, pessimistic approaches to offline RL are ubiquitous, as detailed in the paper's related work, and these existing approaches are arguably identical to the paper's proposed approach.
>
> A1: Actually, existing pessimistic approaches, as listed in the related works, are different from ours.
>
> The main reason is that they consider only the standard MDP or its simple variants where the only goal is to minimize the accumulative cost (or to maximize the accumulative reward): $\min_{\pi}{V}^\pi_1(\underline{s})$​. In this case, the pessimistic approach serves as a scalar penalty. However, in CMOMDPs, the objective is: $\min_{\pi}g(\boldsymbol{V}_1^\pi(\underline s))$​ s.t. $\boldsymbol{V}^\pi_1(\underline{s})\in\mathcal{W}^*$​, which differs from standard MDP in three aspects: (i) the vector-valued cost, (ii) the existence of a preference function $g$​, and (iii) the possibly complex constraints. Due to these reasons, the standard pessimistic approach and theory cannot apply, and how to extend the existing pessimistic approach to this setting has not been studied in existing works. Moreover, these pessimistic works mostly impose strong assumptions such as uniform coverage, while ours do not require it. Therefore, the extended pessimistic approach for CMOMDPs is a significant novelty.
>
> > Q2: ...primal-dual approaches have been proposed for constrained RL...
>
> A2: The primal-dual approach is a common technique in CMDPs. However, in CMOMDPs, the problem is much more difficult, especially when considering nonlinear constraints and preference functions, and the offline settings also add to the difficulty. That is very different from offline CMDPs, and thus existing primal-dual approaches alone cannot apply. Moreover, to the best of our knowledge, even for CMDPs, we are the first to propose a provably efficient offline algorithm that does not require any assumptions on the coverage of the dataset.
>
> > Q3: ... the CMOMDP is essentially a combination of C(onstrained)MDPs and M(ulti)O(bjective)MDPs ...
>
> A3: The CMOMDP is not a simple combination of CMDP and MOMDP. Actually, it is much more difficult. The reason is that, in CMOMDPs, we gain more freedom for the formulation of preference functions and constraints. To see this, we consider two aspects: (i) in existing works of CMDPs, the constraints are usually assumed to be linear (e.g., $\boldsymbol{V}^\pi\le\boldsymbol{b}$​), while we allow nonlinear constraints, and (ii) in previous works of MOMDPs, the preference functions are usually simple and linear, while we consider more complex functions which may take various forms. Moreover, existing works of these weaker problems (CMDPs, MOMDPs) usually require strong coverage assumptions on the dataset (e.g., finite uniform concentrability coefficient), while ours do not impose them. To the best of our knowledge, even for those weaker problems (CMDPs, MOMDPs), our method is the first provably efficient one that does not require any coverage assumption on the offline dataset.
>
> > Q4: The paper does not include any empirical analysis of the algorithm... the reader is left to assume that the algorithm is exclusively theoretical, with no practical ramifications.
>
> A4: We conduct numerical experiments on a simple CMOMDP, which suggests that our algorithm is not only provably efficient but also applicable. Please see the common response of experiments for details, where we also discuss some other potential practical implementation of our algorithm for application. That basically suggests that the algorithm is not merely theoretical. In the revision, we will conduct experiments in more complex scenarios to validate our theory.

---

> > ### Comment · Reviewer_U6xP · 2021-08-13
> > **Thanks**
> >
> > Thank you for the experimental results. These will surely strengthen the submission.

---

### Official Review · Reviewer_M9Rz · 2021-07-16

**Rating:** 6
**Confidence:** 3

**Summary:**

This work considers constrained multi-objective Markov decision processes (CMOMDPs). The authors propose an algorithm to obtain the optimal policy in the offline setting. Optimality and constraint violation bounds are provided.

**Main Review:**

I have some questions:

- The related work section seems to be missing a large number of recent works in constrained RL, specially those using primal-dual approaches.

- A better motivation of the differences between the authors approach and previous work. It seems like the approach that the authors are taking to CMOMDPs could also be applied to CMDPS. A discussion on this regard would be helpful to the reader

- In Section 3.1, the difference between using a standard primal-dual formulation and the formulation (4) should be better motivated.

- The lack of numerical results seems like a major omission. It is difficult to evaluate the proposed algorithm without experiments nor comparisons with previous art.

**Time Spent Reviewing:**

3

---

> ### Author Response · Authors · 2021-08-10
> **Response**
>
> We'd like to thank you for your readings and valuable comments. Your comments are addressed as follows in detail.
>
> > Q1: The related work section seems to be missing a large number of recent works in constrained RL...
>
> A1: We will add more related works of constrained RL to our paper, especially those using primal-dual methods. Please see the common response of related work for details.
>
> > Q2: A better motivation of the differences between the authors approach and previous work. It seems like the approach that the authors are taking to CMOMDPs could also be applied to CMDPS.
>
> A2: Our formulation of CMOMDPs is different from and harder than CMDPs. In particular, we consider the nonlinear preference function and nonlinear constraints, which differ from CMDPs and necessitate the use of duality, as we did in Section 3.1.
>
> Surely, our method can be directly applied to CMDPs without any obstacles since CMOMDPs generalize CMDPs. We discuss the reduction to CMDPs in Appendix C.2. Our theoretical results (e.g., Theorem 1) still apply.
>
> However, existing works on CMDPs cannot apply to CMOMDPs. The reasons are: (i) the cost is vector-valued and the preference function $g$​​​ can be nonlinear in our setting, (ii) the constraints can be nonlinear and thus more difficult to satisfy, and (iii) we do not impose any assumption on the coverage of the dataset while existing works on CMDP usually do (e.g., the concentration coefficient in Assumption 1 of [1]). This suggests that offline CMOMDPs are strictly harder than offline CMDPs, which necessitate a new method as we have proposed.
>
> Moreover, to the best of our knowledge, even for offline CMDPs, we are the first to propose a provably efficient pessimistic algorithm that does not require any assumptions on the coverage of the dataset.
>
>
> > Q3: ... the difference between using a standard primal-dual formulation and the formulation (4) ...
>
> A3: If we use the standard primal-dual approach, say, we introduce a Lagrangian multiplier $\lambda$​​ to (1), we obtain
>
> $$
> \mathsf{\hat p}^*=\min\_{\pi}\max\_{\lambda}\left(\text{SubOpt}(\pi)+\lambda \text{Violation}(\pi)\right).
> $$
>
> Following the same derivation, as we did in Section 3.1, we get the following primal problem
>
> $$
> \hat{\mathsf{p}}^*=
> \min\_{\mathbf{V}\in\mathcal V}
> \max\_{\lambda}
> \max\_{\mathbf{\alpha},\mathbf{\beta}\in\mathcal B^D}
> \mathcal L(\mathbf{V};\mathbf{\alpha},\mathbf{\beta}, \lambda)=
> \mathbf{\beta}^{\top}\mathbf{V}-g^*(\mathbf{\beta})
> -g\big(\mathbf{V}^\*\_1(\underline{s})\big)
> +
> \lambda\mathbf{\alpha}^{\top}\mathbf{V}-\lambda\max\_{\mathbf{x}\in\mathcal W^*}\mathbf{\alpha}^{\top}\mathbf{x}.
> $$
>
> By strong duality (l.214-216), we can also obtain the dual problem,
>
> $$
> \hat{\mathsf{d}}^*=\max\_{\mathbf{\alpha},\mathbf{\beta}\in\mathcal B^D} \max\_{\lambda} \mathsf D(\mathbf{\alpha},\mathbf{\beta},\lambda)
> \quad\text{where}\quad
> \mathsf D(\mathbf{\alpha},\mathbf{\beta}, \lambda)=\min\_{\mathbf{V}\in\mathcal V}\mathcal L(\mathbf{V}; \mathbf{\alpha},\mathbf{\beta},\lambda).
> $$
>
> From this, we can also derive a primal-dual algorithm that can solve CMOMDP, and it should be very similar to our algorithm, PEDI. The only difference is the extra dual variable $\lambda$​​​, which will be updated together with $\alpha$​​​ and $\beta$​​​. However, the multiplier $\lambda$​​​ is **unnecessary**. The preference function $g$​​​ is assumed to be Lipchitz (as required by things like the effective domain of convex conjugate and projected subgradient method). Thus, by simply setting $\lambda>1$​​​ we do not need to solve the maximization with respect to $\lambda$​​ (see l.201 and Appendix D). In other words, using primal-dual approaches to solve our problem is possible with minor modifications, but it is not necessary.
>
> > Q4: The lack of numerical results seems like a major omission...
>
> A4: As our paper is mainly a theoretical one, the empirical performance of the proposed algorithm is not our main claim. However, to show that our algorithm is not only provably efficient but also applicable, we conduct several experiments where we run our algorithm on a simple CMOMDP. It turns out that our algorithm behaves correctly and converges rapidly. Please see the common response of experiments for details. In the revision, we will conduct experiments in more complex scenarios to validate our theory.
>
> ---
>
> [1] Le, H., Voloshin, C., & Yue, Y. (2019). Batch policy learning under constraints. In *International Conference on Machine Learning* (pp. 3703-3712). PMLR.

---

> > ### Comment · Reviewer_M9Rz · 2021-08-25
> > **Re: Response**
> >
> > Thank for your detailed reply. These additions will surely benefit the paper.

---

### Official Review · Reviewer_NsPb · 2021-07-17

**Rating:** 6
**Confidence:** 2

**Summary:**

The paper proposes and analyzes the setting of off-policy constrained multi-objective Markov decision process.  This is a generalized of the constrained MDP setting.  An algorithm to learn the solution is proposed based on a minimax formulation with a pessimistic penalty.  Theoretical analysis defines upper bounds on the suboptimality of the objective and constraint violation. Efficient learning is also demonstrated, under the assumption that the off-policy data sufficiently covers the trajectories induced by the optimal policy.

**Ethical Concerns:**

None that I could identify.



**Limitations And Societal Impact:**

Not discussed. I don’t anticipate any negative impact.  The work may be able to introduce safety constraints into multi-objective MDPs, which would be valuable.

**Main Review:**

The paper defines the setting of off-policy constrained multi-objective MDPs.  I can imagine what might be use cases for this, but it would be nice to present a brief illustrative case, to avoid confusion about what information belongs in the constraint vs in the objective (this is pretty simple when dealing with standard CMDPs but may be less clear with CMOMDP). This would help motivate the work.

The related work section is reasonably well done, but would benefit form including earlier references on offline RL. The novice reader would be tempted to think this originated in 2019. This survey has many earlier references: https://arxiv.org/pdf/2005.01643.pdf

The technical preliminaries are overall quite clearly and thoroughly described.  A strength of the paper is the interesting derivation of the Pessimistic Dual Iteration approach, which (as far as I know) is quite different from the standard CMDP approach (which are more based on adding Lipschitz constraints).   The derivation is quite dense, and the reader needs to be very attentive to follow the notation and construction of the algorithms.  The strong model-based approach (l.222-223) may be a limitation of the method; I see the value for the theoretical analysis, but it would be interesting to see how that plays out in practice.

The theoretical proofs are all pushed to the appendix, and I did not verify them.  The paper would be improved by the additional of remarks to give more intuition on key steps of the proof.  It’s likely the paper would be better suited by a journal format, where proofs can be incorporated in the main body, and reviewers have several more weeks to evaluate the work.

Another limitation of the paper is the complete lack of any empirical results, even a very simple demonstration, to verify the most basic behavior of the algorithm.  This was somewhat disappointing. There is value to just having an implementation as a proof of concept, even if that is not the main claim of the work.

Minor comments:
-	The Slater’s conditions is referred to in several places but not defined.
-	L.84 says “without any assumptions on the coverage of the dataset”, but l.81 (and theoretical analysis later) say  that the analysis assumes that “the trajectories are sufficiently covered”.  Be careful when stating assumptions to avoid confusion about the claims.


**Time Spent Reviewing:**

3 hours

---

> ### Author Response · Authors · 2021-08-10
> **Response**
>
> Thanks for the helpful comments! We will add more early related works to the revision. Please see the common response of related work for details. Illustrative examples and remarks/sketches of proof will also be added, which will give more intuition of our idea. Comments are addressed in detail as follows.
>
> > Q1: The strong model-based approach (l.222-223) may be a limitation of the method.
>
> A1: The model-based approach is not a must for the algorithm.
>
> Recall that by strong duality, we have converted the original problem into its dual version (see l.214-216),
>
> $$
> \mathsf{d}^*=\max\_{\mathbf{\alpha},\mathbf{\beta}\in\mathcal B^D} \mathsf D(\mathbf{\alpha},\mathbf{\beta})
> \quad\text{where}\quad
> \mathsf D(\mathbf{\alpha},\mathbf{\beta})=\min\_{\mathbf{V}\in\mathcal V}\mathcal L(\mathbf{V}; \mathbf{\alpha},\mathbf{\beta}).
> $$
>
> where we have
>
> $$
> \mathcal L(\mathbf {V};\mathbf {\alpha},\mathbf {\beta})=
>     \mathbf {\beta}^{\top}\mathbf {V}-g^*(\mathbf {\beta})
>     -g\big(\mathbf {V}^*_1(\underline{s})\big)
>     +
>     \nu\mathbf {\alpha}^{\top}\mathbf {V}-\nu\max\_{\mathbf {x}\in\mathcal W^*}\mathbf {\alpha}^{\top}\mathbf {x}.
> $$
>
> Essentially, given dual variables $\alpha$ and $\beta$, what method is used to solve the minimization subproblem $\mathsf D(\boldsymbol{\alpha},\boldsymbol{\beta})$ (with either finite or infinite horizon) does not matter, and what we need is only the minimizer (also the value function) $\boldsymbol{V}^\dagger(\alpha,\beta):=\arg\min\_{\mathbf{V}\in\mathcal V}\mathcal L(\mathbf{V}; \mathbf{\alpha},\mathbf{\beta})$, which is used in the dual update (see eq. (9) in l. 240). We can use any method as long as it outputs a pessimistic estimation of $\boldsymbol{V}^\dagger(\alpha,\beta)$ with a theoretical guarantee. That means we can replace the $\text{PessPlanning}$​​ used in Line 5 of Algorithm 2 by any reasonable algorithm that is unnecessarily a model-based one.
>
> > Q2: The paper would be improved by the additional of remarks to give more intuition on key steps of the proof.
>
> A2: We will add the proof sketch to the revised paper, which will better explain the intuition of the proof. The sketch is given below.
>
> **Proof Sketch of Theorem 1.** The proof is done in two steps.
>
> - *First step: upper bounding $\operatorname{SubOpt}(\hat\pi)+\nu \operatorname{Violation}(\hat\pi)$​​​​​​​, which is our objective (see eq. (4)).* Here $\hat\pi$​ is the output of PEDI (Algorithm 2). By pessimism, we have $\operatorname{SubOpt}(\hat\pi)+\nu \operatorname{Violation}(\hat\pi)
> \le
> {g\left(\widehat{\boldsymbol{V}}\_{1}(\underline{s})\right)-g\left(\boldsymbol{V}\_{1}^{\*}(\underline{s})\right)+\nu \operatorname{dist}\left(\widehat{\boldsymbol{V}}\_{1}(\underline{s}), \mathcal{W}^{\*}\right)}$​ where $\widehat{\boldsymbol{V}}\_1(\underline{s})=\frac{1}{K}\sum\_{k=1}^K \boldsymbol{V}\_1^k(\underline{s})$​​. We do linearization via convex conjugate (see eq.(6)) and then get
>
> $$
> \operatorname{SubOpt}(\hat\pi)+\nu \operatorname{Violation}(\hat\pi)
> \le
> K^{-1}\max \_{\|\boldsymbol{\beta}\| \leq 1}\left\\{\boldsymbol{\beta} \cdot \sum\_{k=1}^{K} \boldsymbol{V}\_{1}^{k}(\underline{s})-\sum\_{k=1}^{K} g^{\*}(\boldsymbol{\beta})\right\\}- g\left(\boldsymbol{V}\_{1}^{\*}(\underline{s})\right)
> +K^{-1}\nu \max \_{\|\boldsymbol{\alpha}\| \leq 1}\left\\{\boldsymbol{\alpha} \cdot \sum\_{k=1}^{K} \boldsymbol{V}\_{1}^{k}(\underline{s})-\sum\_{k=1}^{K} \max \_{\boldsymbol{x} \in \mathcal{W}^{\*}} \boldsymbol{\alpha} \cdot \boldsymbol{x}\right\\}.
> $$
>
> Recall that PEDI uses the projected subgradient method, which approximates the max operator above by a sequence of variables at the expense of a sublinear approximation error, which leads to
>
> $$
> \operatorname{SubOpt}(\hat\pi)+\nu \operatorname{Violation}(\hat\pi)
> \leq  K^{-1}\sum\_{k=1}^{K}\left\\{\left(\boldsymbol{\beta}^{k}\right)^{\top} \boldsymbol{V}\_{1}^{k}(\underline{s})-g^{\*}\left(\boldsymbol{\beta}^{k}\right)\right\\}- g\left(\boldsymbol{V}\_{1}^{\*}(\underline{s})\right)
> +K^{-1}\nu \sum\_{k=1}^{K}\left\\{\left(\boldsymbol{\alpha}^{k}\right)^{\top} \boldsymbol{V}\_{1}^{k}(\underline{s})-\max \_{\boldsymbol{x} \in \mathcal{W}^{\*}}\left(\boldsymbol\alpha^{k}\right)^{\top} \boldsymbol{x}\right\\}
> +C(1+\nu) \sqrt{D H^{2}/K}
> $$
>
> where $C$ is a constant. We rearrange the right-hand side above with some tricks and then obtain
>
> $$
> \operatorname{SubOpt}(\hat\pi)+\nu \operatorname{Violation}(\hat\pi)
> \leq
> K^{-1}\sum\_{k=1}^{K}\left[\left(\boldsymbol{\theta}^{k}\right)^{\top}\left(\boldsymbol{V}\_{1}^{k}(\underline{s})-\boldsymbol{V}\_{1}^{\*}(\underline{s})\right)\right]+C(1+\nu) \sqrt{D H^{2}/K}.
> $$
>
> The last term in the right hand side above is exactly $\epsilon\_K$​. Intuitively, the term $\left(\boldsymbol{\theta}^{k}\right)^{\top}\left(\boldsymbol{V}\_{1}^{k}(\underline{s})-\boldsymbol{V}\_{1}^{\*}(\underline{s})\right)$​ indicates the projected difference of state-value functions along $\boldsymbol\theta$​, which is the motivation of $\text{PessPlanning}$​ (Algorithm 1). We apply results from this pessimistic approach and then get the desired conclusion
>
> $$
> \operatorname{SubOpt}(\hat\pi)+\nu \operatorname{Violation}(\hat\pi)
> =
> g\left(\boldsymbol{V}\_{1}^{\widehat{\pi}}(\underline{s})\right)-g\left(\boldsymbol{V}\_{1}^{\*}(\underline{s})\right)+\nu \operatorname{dist}\left(\boldsymbol{V}\_{1}^{\widehat{\pi}}(\underline{s}), \mathcal{W}^{\*}\right)
> \le
> \epsilon\_{K}+\operatorname{IntUncert}\_{\mathcal{D}}^{\pi^{\*}}.
> $$
>
>
> - *Second step: upper bounding $\operatorname{SubOpt}(\hat\pi)$​​​ and $\operatorname{Violation}(\hat\pi)$​​​ individually.* This is done by two facts: (i) $\operatorname{dist}\left(\cdot,\cdot\right) \geq 0$​​​, and (ii) $g(\boldsymbol{V}\_{1}^{\widehat{\pi}}(\underline{s}))-g\left(\boldsymbol{V}\_{1}^{\*}(\underline{s}\right)) \geq-\nu\operatorname{dist}(\boldsymbol{V}\_{1}^{\widehat{\pi}}(\underline{s}),\boldsymbol{V}\_{1}^{\*}(\underline{s})) / 2$​​​​​​. We plug them into the resulting inequality in the first step and then complete the proof.
>
> > Q3: ... lack of any empirical results, even a very simple demonstration, to verify the most basic behavior of the algorithm.
>
> A3: We provide a simple empirical demonstration for our algorithm, which justifies the behavior of our algorithm.  See the common response of experiments for details. In the revision, we will conduct experiments in more complex scenarios to validate our theory.
>
> >Q4: The Slater's conditions is referred to in several places but not defined.
>
> A4: For CMDPs with constraints like $\boldsymbol{V}^\pi\le \boldsymbol{b}$​​, the Slater's condition assumes the existence of a policy $\pi$​​ such that $\boldsymbol{V}^\pi < \boldsymbol{b}$​​. This assumption is common in works regarding CMDPs, such as [1], [2]. We will add this definition to our revised paper.
>
> > Q5: L.84 says ..., but l.81 (and theoretical analysis later) say that ...
>
> A5: In l.84, "without any assumptions on the coverage of the dataset" means our main results (Theorem 1, 2) do not rely on any coverage assumption (e.g., concentrability). In l.81, "the trajectories are sufficiently covered" means that when we have an extra coverage assumption (e.g., concentrability), we can get a further conclusion (Corollary 1), which is derived from Theorem 2 plus the coverage assumption. We will make these statements more clear in the revision.
>
> ---
>
> [1] Qiu, S., Wei, X., Yang, Z., Ye, J., & Wang, Z. (2020). Upper confidence primal-dual reinforcement learning for cmdp with adversarial loss. *arXiv preprint arXiv:2003.00660*.
>
> [2] Efroni, Y., Mannor, S., & Pirotta, M. (2020). Exploration-exploitation in constrained mdps. *arXiv preprint arXiv:2003.02189*.

---

### Official Review · Reviewer_6gH1 · 2021-07-18

**Rating:** 7
**Confidence:** 3

**Summary:**

This paper considers the problem of offline RL in a constrained multi-objective MDP (CMOMDP) environment, where the goal is to compute a policy that achieves the best performance with respect to the given preference function while satisfying the given cost constraints, only using a fixed offline dataset. Pessimism Dual Iteration (PEDI) is proposed, which is the first provably efficient offline RL algorithm for CMOMDP that exploits dual gradient ascent and does not make any assumption on dataset coverage. First, the constrained optimization problem is converted into its dual problem by the convex conjugate. Then, for fixed dual variables, the overall optimization is reduced to an unconstrained planning problem, where pessimistic planning that penalizes uncertainty for each state-action is performed. The dual variables are updated via projected subgradients that are computed by the value of primal variable obtained during planning. Finally, the output policy is a uniform mixture of policies that are obtained during overall iterations. An instantiation of PEDI for linear kernel CMOMDP is introduced. Theoretical results are presented, which bound the suboptimality and constraint violation, and it is minimax optimal for linear kernel CMOMDPs.



**Ethical Concerns:**

-

**Limitations And Societal Impact:**

It seems that the limitations of the work are not explicitly discussed, but I didn't find severe flaws in this work.

**Main Review:**

Recently, offline RL has been getting attention thanks to its potential applicability in real-world situations, but most of them focus on a single-objective RL problem. Still, many practical problems involve multi-objectiveness and constraints, where CMOMDP formulation, a generalization of CMDP,  can be more natural. This work provides the first provably efficient offline RL algorithm for constrained multi-objective problems, which is a valuable contribution. I didn't go through the proof in detail, but the statements look overall sounded.
- Can the analysis and the algorithm be extended to discounted infinite horizon setting? In an infinite-horizon discounted CMDP, there always exists an optimal policy, which is Markov and stationary, but generally stochastic. Can PEDI be also adapted to yield a Markov, stationary, stochastic policy?
- In constrained (MO)MDPs, the constraints are given as hard constraints, thus reducing violations should be the primary objective than reducing suboptimality. Would pessimism alone be sufficient for reducing the chance of cost violation when deployed to the true environment? Are there any further considerations for the robustness of constraint satisfaction?


**Time Spent Reviewing:**

5

---

> ### Author Response · Authors · 2021-08-10
> **Response**
>
> Thank you for your time and feedback! We address your comments in detail as follows.
>
> > Q1: Can the analysis and the algorithm be extended to discounted infinite horizon setting? In an infinite-horizon discounted CMDP, there always exists an optimal policy, which is Markov and stationary, but generally stochastic. Can PEDI be also adapted to yield a Markov, stationary, stochastic policy?
>
> A1: Yes. Our algorithm, PEDI, can be naturally adapted to CMOMDPs (and CMDPs) with discounted infinite horizon.
>
> Recall that by strong duality, we have converted the original problem into its dual version (see l.214-216),
>
> $$
> \mathsf{d}^*=\max\_{\mathbf{\alpha},\mathbf{\beta}\in\mathcal B^D} \mathsf D(\mathbf{\alpha},\mathbf{\beta})
> \quad\text{where}\quad
> \mathsf D(\mathbf{\alpha},\mathbf{\beta})=\min\_{\mathbf{V}\in\mathcal V}\mathcal L(\mathbf{V}; \mathbf{\alpha},\mathbf{\beta}).
> $$
>
> where we have
>
> $$
> \mathcal L(\mathbf {V};\mathbf {\alpha},\mathbf {\beta})=
> 	    \mathbf {\beta}^{\top}\mathbf {V}-g^*(\mathbf {\beta})
> 	    -g\big(\mathbf {V}^*_1(\underline{s})\big)
> 	    +
> 	    \nu\mathbf {\alpha}^{\top}\mathbf {V}-\nu\max\_{\mathbf {x}\in\mathcal W^*}\mathbf {\alpha}^{\top}\mathbf {x}.
> $$
>
> Essentially, given dual variables $\alpha$​​​​ and $\beta$​​​​, what method is used to solve the minimization subproblem $\mathsf D(\boldsymbol{\alpha},\boldsymbol{\beta})$​​​​ (with either finite or infinite horizon) does not matter, and what we need is only the minimizer (also the value function) $\boldsymbol{V}^\dagger(\alpha,\beta):=\arg\min\_{\mathbf{V}\in\mathcal V}\mathcal L(\mathbf{V}; \mathbf{\alpha},\mathbf{\beta})$​​​​, which is used in the dual update (see eq. (9) in l. 240). We can use any method as long as it outputs a pessimistic estimation of $\boldsymbol{V}^\dagger(\alpha,\beta)$​​​​ with a theoretical guarantee. This means that we can replace the $\text{PessPlanning}$​​​​ used in Line 5 of Algorithm 2 by any reasonable algorithm. For instance, for infinite horizon, we can run value iteration till convergence or apply incremental policy optimization, which iteratively updates the policy by $\pi\_{t+1}(a\,|\,s)\propto\pi\_{t}(a\,|\,s)\exp\left(\eta A^{\pi\_t}\_\mathcal P(s,a)\right)$​​​​ where $A$​​​​ is the advantage function for given transition kernel $\mathcal P$​​​​. Moreover, algorithms based on confidence sets are also worth considering (e.g., [2]). These methods may bring extra approximation error as they usually rely on inner iteration, which is usually inevitable in MDPs with infinite horizon, but this error vanishes when the number of inner iteration approaches infinity. Apart from this additional approximation error, the coefficient $H$​​ and the summation $\sum\_{h=1}^H$​​ in Theorem 1 (and also other places) will be replaced by $\mathcal{O}(1/(1-\gamma))$​​​​ if handled properly. All other things will remain unchanged. Although the output policy $\hat\pi$​ is still a mixed policy, it is nearly Markov. The reason is that, when executed, the only non-Markov part of $\hat\pi$​​ happens before the interaction (see l.249).
>
> > Q2: Would pessimism alone be sufficient for reducing the chance of cost violation when deployed to the true environment?
>
> A2: Recall that we have two performance metrics, the suboptimality and constraint violation (see l.171), which may be competing with each other. The role of pessimism in our algorithm is not to optimize them individually but to overcome the spurious correlation of the offline dataset and the algorithm (as stated in Appendix F). The constraint violation depends on the quality of the dataset (as stated in Theorem 1), which is not what pessimism can control. When the dataset has sufficient coverage, constraint violation should be small, as stated in Corollary 1. For specific methods to reduce the constraint violation, please see Q3 and A3 below.
>
> > Q3: Are there any further considerations for the robustness of constraint satisfaction?
>
> A3: Generally speaking, to avoid constraint violation, we can consider the following methods.
>
> **(1) Data Coverage.** As shown in Theorem 1, when the dataset sufficiently covers the trajectories induced by the optimal policy $\pi^*$​​​, the term $\text{IntUncert}^{\pi^*}\_{\mathcal{D}}$​​​​​​​​​ (Theorem 1) will be very small, leading to a small constraint violation of order $\mathcal O(1/\sqrt N)$​. For application, sufficient data coverage can be achieved when the data collecting process is explorative (e.g., the algorithm in [3]).
>
> **(2) Being Conservative.** When deployed to the real environment, we may "shrink" the target set, i.e., making the target set $\mathcal{W}^*$​​​​​ smaller.  This makes the constraint violation more intolerable for our algorithm and thus adds to the robustness. For instance, for constraints like $\boldsymbol{V}^\pi\le \boldsymbol{b}$​​​​​, we can replace it by  $\boldsymbol{V}^\pi\le(1-\epsilon)\boldsymbol{b}$​​​​​ where $0<\epsilon<1$​​​​​. Then if the learned policy $\pi$​​ achieves an $\epsilon\boldsymbol b$​​ violation in $\boldsymbol{V}^\pi\le(1-\epsilon)\boldsymbol{b}$​​, we see that $\boldsymbol{V}^\pi\le\boldsymbol{b}$​​​​​​ holds, and thus the hard constraints are satisfied. An analogue of this idea is studied in [1], in which they have proposed the *Knapsack setting* where the learner wishes to maximize its aggregate reward while respecting these hard constraints.
>
> ---
>
> [1] Brantley, K., Dudik, M., Lykouris, T., Miryoosefi, S., Simchowitz, M., Slivkins, A., and Sun, W. (2020). Constrained episodic reinforcement learning in concave-convex and knapsack settings. *arXiv preprint arXiv:2006.05051*.
>
> [2] Zhou, D., He, J., & Gu, Q. (2021, July). Provably efficient reinforcement learning for discounted mdps with feature mapping. In *International Conference on Machine Learning* (pp. 12793-12802). PMLR.
>
> [3] Jin, C., Krishnamurthy, A., Simchowitz, M., & Yu, T. (2020). Reward-free exploration for reinforcement learning. In *International Conference on Machine Learning* (pp. 4870-4879). PMLR.

---

### Author Response · Authors · 2021-08-10
**Common Response - Experiments**

We conducted experiments as follows. Our code will be made publicly available then.

Here we consider a tabular CMOMDP for illustration. We set the constraint set as $\mathcal{W}^\*=\\{x\in\mathbb R^D: \\|x\\|\_2\le 1\\}$​​ for simplicity, and one can verify that it satisfies Assumption 1 in the paper. The transition kernel $\mathcal{P}$​​ and cost function $\boldsymbol{c}$​​ are generated uniformly at random from $[0,1]$​​ (and we conduct normalization for $\mathcal{P}$​​). We make the cost deterministic for simplicity. In addition, we set $\mathcal{P}\_h(s\_0|s\_0,a\_0)=1$​​ and $\boldsymbol{c}\_h(s\_0,a\_0)=0$​​ for a certain state action pair $(s\_0, a\_0)\in\mathcal S\times\mathcal A$​​ for all $h\in[H]$​​, and the initial state is set to $s\_0$​​. The intuition here is to ensure that the optimal policy, which always takes action $a\_0$​​, achieves zero total cost and zero constraint violation for simplicity. The dataset is generated by a uniformly random experimenter, i.e., it picks $a\in\mathcal{A}$​​ uniformly at random at each step. Hyperparameters are listed below.

| Hyperparameter                               | Value |
| -------------------------------------------- | ----- |
| $H$: horizon                                 | 5     |
| $\|\mathcal{S}\|$: cardinality of state space  | 5     |
| $\|\mathcal{A}\|$: cardinality of action space | 5     |
| $D$: dimension of cost function              | 6     |
| $\|\mathcal{D}\|$: dataset size                | 50000 |
| $K$: number of iteration of PEDI             | 100   |
| $\delta$: confidence level                   | 0.9   |
| $\eta$: step length                          | 0.01  |
| $\nu$: scaling constant                      | 3     |

In our implementation, PEDI estimates the transition and cost functions by the empirical mean, i.e., $\widehat{\mathcal{P}}\_h(s,a)=n\_h(s,a,s')/n\_h(s,a)$ and $\widehat{c}\_h^i(s,a)=f\_h^i(s,a)/n\_h(s,a)$ for $i\in[D]$ where $n\_h(s,a)$ is the number of visits to $(s,a)$ at step $h$ and $f\_h^i(s,a)$ is the sum of the $i$-th cost incurred in the dataset when visiting $(s,a)$ at step $h$. We construct the Hoeffding-style uncertainty quantifiers, i.e., $\Gamma^{\mathcal{P}}\_h(s,a,s')=\sqrt{\log(2 H|\mathcal{S}||\mathcal{A}||\mathcal{S}|/\delta)/(2n\_h(s,a))}$ and $\Gamma\_h^{\boldsymbol{c}}=\sqrt{\log(2DH|\mathcal{S}||\mathcal{A}|/\delta)/(2n\_h(s,a))}$. We can verify that they satisfy the definition (Definition 1).

We conduct experiments to see whether PEDI converges to the optimal policy in two different preference functions.

**(1) Quadratic Functions.**  Suppose the interplay of cost functions can be modeled by a positive definite matrix $A$​​​​​​​, a vector $b$​​​​​​​​ and a constant $c$, i.e., the preference function is defined as

$$
g(x)=\frac{1}{2}x^\top Ax+b^\top x+c
$$

where $A$​ is positive definite. For simplicity, we assume $b$​ is the zero vector and $c=0$​. To guarantee 1-Lipschitzness, it suffices to restrict the spectral radius $\lambda\_{\max}$. In particular, we require $\lambda\_{\max}(A)\le 1/(2HD^{1/2})$ since $\\|\partial\_x g\\|\_2=\\|2A x\\|\_2\le 2\lambda\_{\max}(A)HD^{1/2}$. For the convex conjugate, we can verify that $g^\*(x^\*)=\frac{1}{2}(x^\*-b)^\top A^{-1} (x^\*-b)-c=\frac{1}{2}x^\*A^{-1}x^\*$. In the numerical experiment, the matrix $A$​ is randomly generated with the mentioned spectral radius requirement. The results are given below.

| Iteration $k$ | Suboptimality | Constraint Violation |
| ------------- | ------------- | -------------------- |
| 1             | 0.067         | 0.880                |
| 2             | 0.505      | 4.007            |
| 3             | 0.067         | 0.880                |
| 4 $\sim$​ 100            | 0.000       | 0.000              |

As we see, it converges to the optimal policy after mere four iterations and stays optimal permanently.

**(2) Polynomial Functions.** Suppose the preference function is polynomial, i.e.,

$$
g(x)=\sum\_{i=1}^D c\_i |x\_i|^{p\_i}.
$$

For simplicity, we assume $p=p\_i=p\_j$​ and $c=c\_i=c\_j$​ for any $1\le i,j\le D$​. To ensure 1-Lipschitzness, it suffices to set $c=1/(pH^{p-1} D^{1/2})$​ for all $i$​ which results in $\\|\partial\_x g\\|\_2=c p x^{p-1}D^{1/2}\le 1$​ for $x\ge 0$​. Then, we have $g^\*(x^\*)=\sum\_{i=1}^D\frac{|x^\*\_i|^q}{c^{q-1}p^{q-1}q}$​ where $\frac{1}{p}+\frac{1}{q}=1$​. In the numerical experiment, we set $p=2$​. The results are given below.

| Iteration $k$ | Suboptimality | Constraint Violation |
| ------------- | ------------- | -------------------- |
| 1             | 0.165         | 1.009                |
| 2             | 0.139         | 0.844                |
| 3 $\sim$ 100  | 0.000         | 0.000                |

As we see, it reaches the optimal solution after only three iterations.

The following two remarks discuss (1) possibilities to handle other (even more general) preference functions and (2) the possibilities of some practical variants of PEDI for application, which is left to future work, as this paper is mainly a theoretical one.

**Remark 1. (General Preference Function)** In addition to the above demonstration, PEDI is also easily applicable to preference functions from many function classes such as exponential function, logarithmic function, and entropy function. Even when the exact expression of the preference function $g$​​​​ is not good or even unknown, PEDI applies as long as we can approximate $g^\*$​​​​ by some numerical methods, say, by directly approximate $\sup\_x (\langle x^\*,x\rangle-g(x))$​​​​​​​, which is the definition of the convex conjugate. To obtain the subgradient, we can use certain techniques such as numerical differentiation.

**Remark 2. (General Planning Algorithm)** For a real-world application, the pessimistic planning ($\text{PessPlanning}$​​, see Algorithm 1) may seem too heavy. It can be replaced by any algorithms as long as it approximately outputs the desired policy $\pi^k$​​ and a pessimistic estimation of the value functions $V^k$​​​ at each iteration. For example, we can apply policy iteration algorithms or even any neural network-based approximate algorithms.

---

### Author Response · Authors · 2021-08-10
**Common Response - Related Work**

**Early Reference of Offline RL (For Reviewer NsPb).** We will integrate the following references into the related work section additionally.

[1], [2] are tutorials and surveys of offline RL.

[3] studies the necessary conditions that permit provable sample-efficient algorithm and emphasize the significant influence of insufficient coverage of the dataset, which is also reflected in our main results (e.g., Theorem 1).

[17], [18], [6] establish theoretical guarantees for offline RL based on approximate dynamic programming. Nevertheless, they do not take constraints into account, thus failing to handle constrained RL.

[4], [5], [6], [7], [8]  assume concentrability coefficient to be upper bounded, which connect to Corollary 1. However, our results apply even when we do not have those assumptions.

**More Recent Works in Constrained RL (For Reviewer M9Rz).** The following works will be integrated into the related work section.

[9] is a comprehensive tutorial of CMDPs.

[12], [14], [15]  study constrained RL with simple linear constraints, and [10], [11], [13] consider RL with more complex constraints. All of them utilize the primal-dual approach, which alone will not be sufficient to solve CMOMDPs. Moreover, they largely consider online RL, which follows a different analysis compared with offline RL.

[16] consider offline CMDP and propose algorithms with provable guarantees. Nevertheless, they impose the concentrability assumption (Assumption 1). Although this assumption limits the severity of distribution shift, it often fails to hold in practice. However, we do not need this assumption, and thus ours is more practical than theirs even when we reduce our method to offline CMDPs.

---

[1] Lange, S., Gabel, T. and Treadmiller, M. (2012). Batch reinforcement learning. In Reinforcement learning. Springer, 45–73.

[2] Levine, S., Kumar, A., Tucker, G. and Fu, J. (2020). Offline reinforcement learning: Tutorial, review, and perspectives on open problems. arXiv preprint arXiv:2005.01643.

[3] Wang, R., Foster, D. P. and Kakade, S. M. (2020). What are the statistical limits of offline RL with linear function approximation? arXiv preprint arXiv:2010.11895.

[4] Antos, A., Szepesv´ari, C. and Munos, R. (2007). Fitted Q-iteration in continuous action-space MDPs. In Advances in Neural Information Processing Systems.

[5] Antos, A., Szepesv´ari, C. and Munos, R. (2008). Learning near-optimal policies with Bellman- residual minimization based fitted policy iteration and a single sample path. Machine Learning, 71 89–129.

[6]Farahmand, A.-m., Szepesv'ari, C. and Munos, R. (2010). Error propagation for approximate policy and value iteration. In Advances in Neural Information Processing Systems.

[7] Fu, Z., Yang, Z. and Wang, Z. (2020). Single-timescale actor-critic provably finds globally optimal policy. arXiv preprint arXiv:2008.00483.

[8] Scherrer, B., Ghavamzadeh, M., Gabillon, V., Lesner, B. and Geist, M. (2015). Approximate mod- ified policy iteration and its application to the game of Tetris. Journal of Machine Learning Research, 16 1629–1676.

[9] Altman, E. Constrained Markov decision processes, volume 7. CRC Press, 1999.

[10] Bhatnagar, S., & Lakshmanan, K. (2012). An online actor–critic algorithm with function approximation for constrained markov decision processes. *Journal of Optimization Theory and Applications*, *153*(3), 688-708.

[11] Chow, Y., Ghavamzadeh, M., Janson, L., & Pavone, M. (2017). Risk-constrained reinforcement learning with percentile risk criteria. *The Journal of Machine Learning Research*, *18*(1), 6070-6120.

[12] Tessler, C., Mankowitz, D. J., & Mannor, S. (2018). Reward constrained policy optimization. *arXiv preprint arXiv:1805.11074*.

[13] Paternain, S., Calvo-Fullana, M., Chamon, L. F., & Ribeiro, A. (2019). Safe policies for reinforcement learning via primal-dual methods. *arXiv preprint arXiv:1911.09101*.

[14] Ding, D., Wei, X., Yang, Z., Wang, Z., & Jovanovic, M. (2021). Provably efficient safe exploration via primal-dual policy optimization. In *International Conference on Artificial Intelligence and Statistics* (pp. 3304-3312). PMLR.

[15] Qiu, S., Wei, X., Yang, Z., Ye, J., & Wang, Z. (2020). Upper confidence primal-dual reinforcement learning for cmdp with adversarial loss. *arXiv preprint arXiv:2003.00660*.

[16] Le, H., Voloshin, C., & Yue, Y. (2019, May). Batch policy learning under constraints. In *International Conference on Machine Learning* (pp. 3703-3712). PMLR.

[17] Tosatto, S., Pirotta, M., d’Eramo, C., & Restelli, M. (2017, July). Boosted fitted q-iteration. In *International Conference on Machine Learning* (pp. 3434-3443). PMLR.

[18] Farahmand, A. M., Ghavamzadeh, M., Szepesvári, C., & Mannor, S. (2016). Regularized policy iteration with nonparametric function spaces. *The Journal of Machine Learning Research*, *17*(1), 4809-4874.

---

### Decision · Program_Chairs · 2021-09-27

**Decision:**

Accept (Poster)

**Comment:**


In this paper, the authors consider the offline learning for constrained multi-objective MDP (CMOMDP). Specifically, the authors proposed an algorithm which exploits the primal dual structure with pessimistic planning. The algorithm is extended for linear kernel CMOMDP. Moreover, the authors also provided rigorous suboptimality and constraint violation guarantees.

Although as most of the reviewers pointed out, the original version of the paper is lacking some related work discussion about offline RL and constrained RL, and empirical justification of the algorithm, the authors provided these parts during the rebuttal period, therefore,  making the paper more convincing. I recommend acceptance for this paper. Please take the reviewers's other suggestions, especially adding the intuition/sketch of the proof into main text, to improve the paper.